

Atmospheric
Measurement
Techniques



# Characterization of atmospheric aerosol optical properties based on the combined use of a ground-based Raman lidar and an airborne optical particle counter in the framework of the Hydrological Cycle in the Mediterranean Experiment – Special Observation Period 1

**Dario Stelitano**[1,a], **Paolo Di Girolamo**[1], **Andrea Scoccione**[3,b], **Donato Summa**[2], **and Marco Cacciani**[3]

[1]Scuola di Ingegneria, Università degli Studi della Basilicata, 85100 Potenza, Italy
[2]~~Osservatorio Nazionale Terremoti, Istituto Nazionale di Geofisica e Vulcanologia, 00143 Rome, Italy~~
[3]Dipartimento di Fisica, Università di Roma "La Sapienza", 00100 Rome, Italy
[a]now at: Osservatorio Nazionale Terremoti, Istituto Nazionale di Geofisica e Vulcanologia, 00143 Rome, Italy
[b]now at: Centro Operativo per la Meteorologia, Aeronautica Militare, 00040 Pomezia, Italy

**Correspondence:** Paolo Di Girolamo (paolo.digirolamo@unibas.it)

**Abstract.** Vertical profiles of the particle backscattering coefficient at 355, 532 and 1064 nm measured by the University of Basilicata Raman lidar system (BASIL) have been compared with simulated particle backscatter profiles obtained through a Mie scattering code based on the use of simultaneous and almost co-located profiles provided by an airborne optical particle counter. Measurements were carried out during dedicated flights of the French research aircraft ATR42 in the framework of the European Facility for Airborne Research (EUFAR) project "WaLiTemp", as part of the Hydrological Cycle in the Mediterranean Experiment – Special Observation Period 1 (HyMeX-SOP1). Results from two selected case studies are reported and discussed in the paper, and a dedicated analysis approach is illustrated and applied to the dataset. Results reveal a good agreement between measured and simulated multi-wavelength particle backscattering profiles. Specifically, simulated and measured particle backscattering profiles at 355 and 532 nm for the second case study are found to deviate less than 15 % (mean value = 7 %) and 50 % (mean value = 30 %), respectively, when considering the presence of a continental–urban aerosol component, while slightly larger deviation values are found for the first study. The reported good agreement between measured and simulated multi-wavelength particle backscatter profiles

testifies to the ability of multi-wavelength Raman lidar systems to infer aerosol types at different altitudes.

## 1 Introduction

Aerosols are a key atmospheric component, playing a major role in meteo-climatic processes. Aerosols influence precipitation processes and the water cycles primary through two effects: the direct effect, as a result of the scattering/absorption of solar radiation (among others, Haywood and Boucher, 2000; Takemura et al., 2005), and the indirect effect, as a result of the interaction with clouds (among others, Sekiguchi et al., 2003; Yang et al., 2011). A semi-direct effect can also arise in the presence of high aerosol loading, determining scattering and absorption enhancement, ultimately leading to an alteration of atmospheric stability (e.g. Mitchell, 1971). Despite the well-recognized aerosol importance in meteorological processes and climate evolution, only a limited number of remote sensing techniques can provide vertically resolved measurements of the microphysical properties of aerosol particles (among others, Bellantone et al., 2008; Granados-Muñoz et al., 2016; Mhawish et al., 2018). For example, in situ sensors transported by aerostatic bal-

loons or any other flying vector allow the vertical profile of aerosol size and microphysical properties to be measured, with high vertical resolution (of the order of 10 m) but typically with a limited temporal resolution. Any experiment aimed at characterizing the temporal evolution of aerosol microphysical properties would require several consecutive balloon launches or flights, with the time lag between two consecutive launches/flights unlikely being shorter than 1 h, with a consequent detriment of the temporal resolution. Additionally, in situ particle sensors are quite heavy and bulky, which – in the case of balloon-borne experiments – implies the use of quite large aerostatic balloons. This makes monitoring by in situ particle sensors very expensive and logistically difficult to implement.

Remote sensing techniques can overcome these limitations. A variety of passive optical remote sensors (i.e. spectroradiometers, sun and sky photometers, etc.) have demonstrated their capability to characterize aerosol microphysical properties, but they lack in vertical resolution, which makes them scarcely suited for vertically resolved measurements of aerosol size and microphysical properties. Low vertical resolution is combined with a limited temporal resolution when these techniques are implemented on sun-synchronous orbiting platforms, with a typical "revisit time" of several hours. Active remote sensing systems may overcome this limitation. Specifically, lidar systems with aerosol measurement capability are characterized by high accuracies and temporal/vertical resolutions, which makes them particularly suited for aerosol typing applications. Lidar measurements of aerosol optical properties have been reported since the early 1960s (among others, Fiocco and Grams, 1964; Elterman, 1966). Originally, measurements were carried out with single-wavelength elastic backscatter lidars capable of providing vertical profiles of the particle backscattering coefficient at the laser wavelength. In these systems the particle backscattering coefficient is determined from the elastic lidar signals based on the application of the Klett–Fernald–Sasano approach (Klett, 1981, 1985; Fernald, 1984) or similar derived approaches (Di Girolamo et al., 1995, 1999). More recently, the acquired capability to measure roto-vibrational Raman lidar echoes from nitrogen and oxygen molecules has made the determination of the particle extinction coefficient also possible (Ansmann et al., 1990, 1992). The possibility of retrieving particle size and microphysical parameters from multi-wavelength lidar data of particle backscattering, extinction and depolarization has been recently demonstrated by a variety of authors (Müller et al., 2001, 2007, 2009; Veselovskii et al., 2002, 2009, 2010). These measurements can be combined with simultaneous measurements of the atmospheric thermodynamic profiles (Wulfmeyer et al., 2005; Di Girolamo et al., 2008, 2018a) to characterize aerosol–cloud interaction mechanisms. The ground-based University of Basilicata Raman lidar system (BASIL) has demonstrated the capability to provide multi-wavelength Raman lidar measurements with high quality and accuracy for the retrieval

of particle size and microphysical parameters (Veselovskii et al., 2010; Di Girolamo et al., 2012a). The system was deployed in Candillargues (southern France) in the period from August to November 2012 in the framework of the Hydrological cycle in the Mediterranean Experiment (HyMeX) Special Observation Period 1 (SOP1). In the present paper, measurements carried out by BASIL are illustrated with the purpose of characterizing atmospheric aerosol optical properties. These measurements, in combination with in situ measurements from an airborne optical particle counter and the application of a Mie scattering code, are used to infer aerosol types. Back-trajectory analyses from a Lagrangian model (HYSPLIT) are used in support of the assessment of aerosol types (Man and Shih, 2001; Methven et al., 2001; Estellés et al., 2007; Toledano et al., 2009). The outline of the paper is as follows: Sect. 2 provides a description of the Raman lidar system BASIL and the airborne optical particle counter; Sect. 3 illustrates HyMeX-SOP1. The methodology is illustrated in Sect. 4, while measurements and simulations are reported in Sect. 5. Finally, Sect. 6 summarizes all results and provides some indications for possible future follow-up activities.

## 2   Instrumental setup

### 2.1   BASIL

The Raman lidar BASIL has been developed around a pulsed Nd:YAG laser, emitting pulses at 355, 532 and 1064 nm, with a repetition rate of 20 Hz. The system includes a large aperture telescope in Newtonian configuration, with a 400 mm diameter primary mirror, primarily aimed at the collection of Raman and higher range signals. Two additional smaller telescopes, developed around two 50 mm diameter 200 mm focal length lenses, are used to collect the backscatter echoes at 1064 nm and the total and cross-polarized backscatter echoes at 532 nm. The laser emission at 355 nm (average power of 10 W) is used to stimulate Raman scattering from water vapour and nitrogen and oxygen molecules (Di Girolamo et al., 2004, 2006, 2009a), which are ultimately used to measure the vertical profiles of atmospheric temperature, water vapour mixing ratio and aerosol extinction coefficient at 355 nm. Elastic backscattering echoes from aerosol and molecular species at 355, 532 and 1064 nm, in combination with the Raman scattering echoes from molecular nitrogen, are used to measure the vertical profiles of the aerosol backscattering coefficient at these three wavelengths. More details of the considered approaches are given in Sect. 4. Raman echoes are very weak and degraded by solar radiation in daytime. Consequently, high laser powers and large aperture telescopes are required to measure daytime Raman signals with a sufficient signal-to-noise ratio throughout a large portion of the troposphere. The instrumental setup of BASIL has been described in detail in several previous papers (Di Giro-

lamo et al., 2009a, b, 2012a, b, 2016, 2017; Bhawar et al., 2011). BASIL was deployed in a variety of international field campaigns (among others, Bhawar et al., 2008; Serio et al., 2008; Wulfmeyer et al., 2008; Bennett et al., 2011; Ducrocq et al., 2014; Macke et al., 2017; Di Girolamo et al., 2018b).

## 2.2 Optical particle counter

An optical particle counter (OPC), manufactured by GRIMM Aerosol Technik GmbH (model Sky-OPC 1.129), is used to measure the size-resolved particle number concentration $dN/dr$ in the size range 0.25–32 μm. The sensor includes 31 size bins. The laser beam generated by a 683 nm diode laser invests the aerosol particles exiting from a pump chamber; the scattered radiation is deflected by two separate mirrors and detected by a photon sensor (Heim et al., 2008). By summing up the particle number over all the size intervals, the total number concentration is derived (Grimm and Eatough, 2009). The OPC model used in the present effort has a specific airborne design (McMeeking et al., 2010). The use of a differential pressure sensor and an external pump allows OPC measurements to be performed independently of environmental pressure conditions. The OPC was installed on board the French research aircraft ATR42, operated by the Service des Avions Instrumentés pour la Recherche en Environnement (SAFIRE), as part of an ensemble of in situ sensors for the characterization of aerosol and cloud size and microphysical properties. Dedicated flights by the ATR42 were performed during HyMeX-SOP 1 in the framework of the European Facility for Airborne Research (EUFAR) project "WaLiTemp", with the aircraft looping up and down in the proximity of the Raman lidar system.

## 3 HyMeX and the Special Observation Period 1

The Hydrological cycle in Mediterranean Experiment was conceived with the overarching goal of collecting a large set of atmospheric and oceanic data to be used to get a better understanding of the hydrological cycle in the Mediterranean area. Within this experiment a major field campaign, the Special Observation Period 1 (SOP1), took place over the north-western Mediterranean area in the period September–November 2012 (Ducrocq et al., 2014). During HyMeX-SOP1 the Raman lidar system BASIL was deployed in the Cévennes-Vivarais atmospheric "supersite", located in Candillargues (43°37′ N, 4°04′ E; elevation: 1 m). BASIL was operated from 5 September to 5 November 2012, collecting more than 600 h of measurements, distributed over 51 measurement days, and consisted of 19 Intensive Observation Periods (IOPs).

The French research aircraft ATR42, hosting the OPC, was stationed at Montpellier Airport. Its main payload consisted of the airborne DIAL LEANDRE 2, profiling water vapour mixing ratio beneath the aircraft. The ATR42 payload also included in situ sensors for turbulence measurements, as well as aerosol and cloud microphysics probes, including the OPC. During HyMeX-SOP1, the ATR42 performed more than 60 flight hours: 8 were supported by the EUFAR project WaLiTemp, and the remaining hours were supported by the "Mediterranean Integrated STudies at Regional and Local Scales" (MISTRALS) programme. A specific flight pattern was defined for the purposes of the WaLiTemp project (Fig. 1), with the aircraft making spirals (hippodromes) up and down around a central location, originally aimed to be the atmospheric supersite in Candillargues. Unfortunately, because of air traffic restrictions, aircraft sensors' operation was typically started 20 km eastward of the supersite, and the central location of the hippodromes was also moved 20 km eastward.. Flights hours in the framework of the WaLiTemp project were carried out on 13 September, 2 and 29 October and 5 November 2012.

Spiral ascents and descents were carried out with a vertical speed of 150 m min$^{-1}$. During each flight, except in the presence of specific logistic issues, a minimum of two ascent–descent spirals were carried out. For the purposes of the present comparisons, in order to minimize the effect associated with the sounding of different air masses, we selected days characterized by horizontally homogeneous atmospheric conditions.

## 4 Methodology

The particle volume backscattering coefficient can be expressed as

$$\beta_{\lambda_0}^{\mathrm{par}} = \int\limits_0^\infty Q_{\mathrm{back}}(r)\, n(r)\, dr, \tag{1}$$

with $Q_{\mathrm{back}}(r)$ being the particle backscattering efficiency and $n'(r) = dN/dr$ being the particle size distribution. $Q_{\mathrm{back}}(r)$ can be expressed as (Grainger et al., 2004)

$$Q_{\mathrm{back}} = \frac{2}{x^2} \sum_{n=1}^\infty (2n+1)\left(|a_n|^2 + |b_n|^2\right), \tag{2}$$

where the terms $a_n$ and $b_n$ represent the Mie scattering amplitudes of the $n$th magnetic partial wave ($n$ being the function order). $a_n$ and $b_n$ are obtained through the following expressions:

$$a_n = \frac{\psi_n(x)\,\psi_n'(mx) - m\psi_n'(x)\,\psi_n(mx)}{\xi_n^{(1)}(x)\,\psi_n'(mx) - m\xi_n^{(1)'}(x)\,\psi_n(mx)} \tag{3}$$

$$b_n = \frac{\psi_n'(x)\,\psi_n(mx) - m\psi_n(x)\,\psi_n'(mx)}{\xi_n^{(1)'}(x)\,\psi_n'(mx) - m\xi_n^{(1)}(x)\,\psi_n(mx)}, \tag{4}$$

where $m$ is the complex refractive index; $x = 2\pi r/\lambda$ is the particle size parameter, with $\lambda$ being the laser wavelength and $r$ being the particle radius, assumed to be a sphere. $\psi_n(x)$

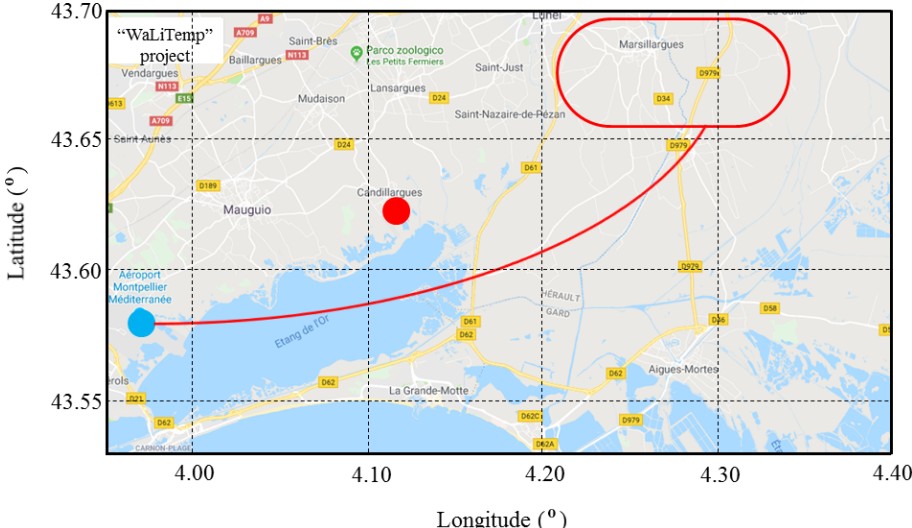

**Figure 1.** ATR42 flight pattern in the framework of the WaLiTemp project (red line). The light blue dot represents the position of Montpellier Airport, where the ATR-42 took off and landed, while the red dot represent the position of the Raman lidar BASIL. The red curve represents the footprint of the aircraft pattern, including the positions of the spirals (hippodromes) up and down and the ground track from the airport to the spiraling position. The distance between the lidar site and the flight pattern is approx. 20 km.

and $\xi_n^{(1)}$ are Riccati–Bessel functions defined in terms of the spherical Bessel function of the first kind (Temme, 1996). A log-normal size distribution is considered in this study, with an analytical expression for each mode of the form (Grainger et al., 2004):

$$ n'(r) = \frac{N_0}{\sqrt{2\pi}} \frac{1}{\ln S} \frac{1}{r} \exp\left[ -\frac{([\ln r - \ln r_m]^2)}{2\ln^2 S} \right], \tag{5} $$

where $n'(r) = \mathrm{d}N/\mathrm{d}r$ is the number of particles within the size interval $\mathrm{d}r$, with $N(r)$ representing the cumulative particle number distribution for particles larger than $R$, $r_m$ is the median radius of the distribution, $S$ is the standard deviation of the distribution and $N_0$ is the particle integral concentration for the considered mode. $S$ is a measure of the particle polydispersity, with $\ln S$ being equal to 1 for monodisperse particles. The log-normal distribution is completely described by $N_0$, $r_m$ and $S$. Three modes are typically considered to describe the different aerosol components (d'Almeida et al., 1991): a fine or nucleation particle mode, a large or accumulation particle mode and a giant or coarse particle mode.

For the purposes of this research effort, particle concentration $N_0$ is obtained by minimizing differences between the size distribution measured by the OPC and the simulated distribution, while the values of $r_m$ and $S$ are those identified in the following section based on literature results. Simulated backscatter profiles $\beta_{\lambda_0}^{\mathrm{par}}(z)$ are obtained through the application Eq. (1) for all altitudes covered by the OPC, considering different refractive index and size parameters' values for the three distribution modes, in dependence of the aerosol type, and integrating the expression over the three distribution modes. To perform these computations a specific Mie scat-

tering code was developed by the authors in an IDL environment. The possibility to retrieve the particle size and microphysical properties from multi-wavelength measurements of the particle backscattering and extinction coefficient has been demonstrated by several authors (among others, Müller et al., 2001; Veselovskii et al., 2002) based on the application of retrieval schemes employing Tikhonov's inversion with regularization, which apply Mie scattering theory to an ensemble of particles with spherical shape. However, an appropriate and effective application of this approach imposes the use of particle backscatter and extinction profiles with a statistical uncertainty not exceeding 5 %–10 %. Multi-wavelength Raman lidar measurements of the particle backscattering and extinction coefficient for the considered case studies were not characterized by such a low level of uncertainty, this being especially true for the particle backscatter measurements at 1064 nm.

In order to determine aerosol typology, deviations between measured and simulated particle backscattering profiles at 355 and 532 nm were minimized. Initial values in terms of modal radius, $\bar{r}$, standard deviation, $\sigma$, and refractive index for the different aerosol components were taken from d'Almeida et al. (1991). At each altitude, the particle size distribution measured by the optical particle counter is compared with the five aerosol typologies listed in d'Almeida et al. (1991), which for the sake of clarity are reproduced below:

– average continental (continental environment influenced by anthropogenic pollution);

– urban (continental environment heavy influenced by anthropogenic pollution);

– maritime polluted (environment polluted as Mediter-
ranean Sea or North Atlantic);

– clean–polar (Arctic environment during summer pe-
riod);

– clean continental–rural (rural continental environment
without pollution).

Specifically, both urban and continental aerosols include a
soot and pollution fine-mode component (as both aerosol
types include the same aerosol components, they are treated
in what follows as a single aerosol type), a water-soluble
accumulation-mode component and a dust-like coarse-mode
component; the maritime polluted aerosol type includes a
soot and pollution fine-mode component, a water-soluble
accumulation-mode component and a sea-salt coarse-mode
component; the summertime Arctic aerosol type includes
a sulfate fine-mode component and a sea salt and mineral
accumulation-mode component; the rural aerosol type in-
cludes a water-soluble accumulation-mode component and
a dust-like coarse-mode component.

D'Almeida et al. (1991), Junge and Jaenicke (1971) and
Junge (1972) suggested the use of a tri-modal log-normal
size distribution (see Eq. 5), indicating specific values for the
two primary size distribution parameters, i.e. the modal ra-
dius, $\bar{r}$, and standard deviation, $\sigma$. Values of the modal radius,
the standard deviation and the real, $n_r$, and imaginary part,
$n_i$, of refractive index at the three lidar wavelengths (355,
532 and 1064 nm) for the three different aerosol components
considered in the present computations are inferred from dif-
ferent papers in the literature (d'Almeida et al., 1991; Shettle
and Fenn, 1976, 1979; WCP–112, 1986) and are listed in Ta-
ble 1.

The log-normal size distribution has been computed con-
sidering the OPC data in the dimensional range 0.25–2.5 µm,
with a 300 m vertical integration window. Results are illus-
trated in Fig. 2 (bold black line). In this same figure the
size distribution computed from the OPC data is compared
with the theoretical distributions for the three different modes
(fine mode – red line, accumulation mode – violet line, coarse
mode – light blue line).

For each of the three modes, the number of particles has
been varied in order for the total theoretical distribution (thin
black line) to match the experimental distribution computed
with the OPC data. The matching between the experimental
and theoretical distributions has been optimized based on the
application of a best fit procedure. This approach was applied
to each altitude level. In Fig. 2, we consider experimental and
theoretical distributions at an altitude of 1529 m, this being
the lowest altitude at which aerosols larger than 0.7–0.8 µm
were measured by the OPC.

The vertical profiles of the particle backscattering coeffi-
cient at 355, 532 and 1064 nm have been simulated through
the above-mentioned Mie scattering code from the OPC data,
considering values of $\bar{r}$ and $\sigma$ for the different aerosol com-

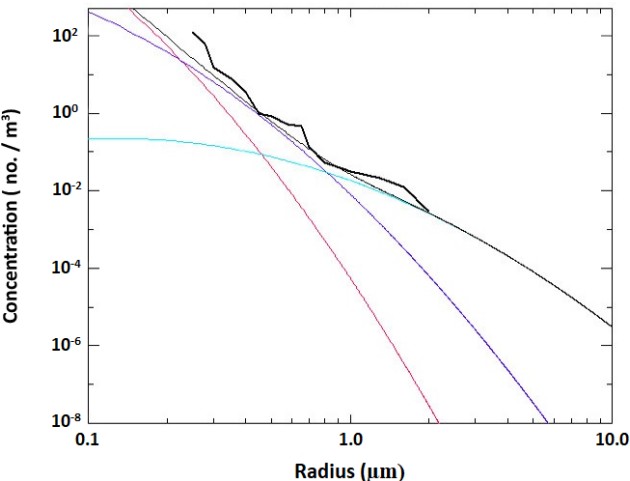

**Figure 2.** Size distribution computed from the OPC data (bold black
line), together with the total theoretical distribution (thin black line)
and theoretical distributions for the three different modes: fine mode
(soot and pollution, red line), accumulation mode (water-soluble
aerosols, violet line) and coarse mode (sea salt, light blue line).

ponents. Measured profiles of the particle backscattering co-
efficient profiles at 355 and 532 nm are obtained from the
Raman lidar signals through the application of the Raman
techniques, which relies on the ratio between the 355/532 nm
elastic signal and the corresponding simultaneous molecu-
lar nitrogen roto-vibrational Raman signal. The two signals
are characterized by an almost identical overlap function,
and therefore the overlap effect is cancelled out when ra-
tioing the signals. Conversely, particle backscattering coeffi-
cient profiles at 1064 nm are obtained through the application
of a Klett-modified inversion approach (Di Girolamo et al.,
1995, 1999). The specific approach used in the present anal-
ysis considers a height-dependent lidar ratio profile and an
iterative procedure converging to a final particle backscatter-
ing profile (Di Girolamo et al. 1995, 1995). Additionally, the
elastic backscatter signal at 1064 nm and an additional elas-
tic backscatter signal at 532 nm are collected with two small
telescopes, developed around two 50 mm diameter 200 mm
focal length lenses, with overlap regions not extending above
3–400 m.

A modified version of the approach defined by Di Iorio
et al. (2003) was applied in order to determine the sounded
aerosol typology. This approach is based on the minimiza-
tion of the relative deviation between the measured and the
simulated particle backscattering coefficient; i.e.

$$\Delta = \frac{1}{N_p} \sum_{k=1}^{N} \frac{\left| \beta_{\lambda\,(\text{simulated})}\,(z_k) - \beta_{k\,(\text{measured})}\,(z_k) \right|}{\beta_{\lambda\,(\text{measured})}(z_k)}, \tag{6}$$

where $z_k$ is the altitude.

In the attempt to simultaneously minimize deviations be-
tween measured and simulated particle backscattering pro-
files at 355, 532 and 1064 nm, a total deviation can be com-

**Table 1.** Modal radius, standard deviation and refractive index (real and imaginary part) for the different considered aerosol components (from d'Almeida et al., 1991).

| | $\overline{r}$ (µm) | $\sigma$ | $m_r$ (355 nm) | $m_i$ (355 nm) | $m_r$ (532 nm) | $m_i$ (532 nm) |
|---|---|---|---|---|---|---|
| Soot | 0.012 | 2.00 | 1.75 | $4.65 \times 10^{-1}$ | 1.75 | $4.44 \times 10^{-1}$ |
| Water-soluble | 0.024 | 2.24 | 1.53 | $5.00 \times 10^{-3}$ | 1.53 | $6.00 \times 10^{-3}$ |
| Dust-like | 0.471 | 2.51 | 1.53 | $8.00 \times 10^{-3}$ | 1.53 | $8.00 \times 10^{-3}$ |
| Sea salt (fine) | 0.300 | 2.51 | 1.39 | $1.20 \times 10^{-7}$ | 1.38 | $3.70 \times 10^{-9}$ |
| Sulfate | 0.069 | 2.03 | 1.45 | $1.00 \times 10^{-8}$ | 1.43 | $1.00 \times 10^{-8}$ |
| Sea salt (acc.) | 0.400 | 2.03 | 1.39 | $1.20 \times 10^{-7}$ | 1.38 | $3.7 \times 10^{-9}$ |
| Mineral | 0.270 | 2.67 | 1.53 | $1.70 \times 10^{-2}$ | 1.53 | $5.50 \times 10^{-3}$ |

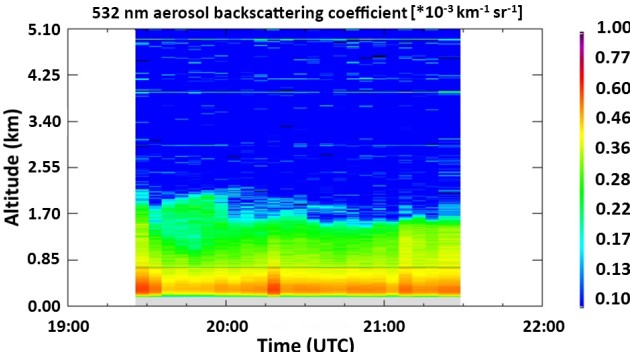

**Figure 3.** Time evolution of the particle backscattering coefficient at 532 nm over the time interval 19:30–21:30 UTC on 13 September 2012.

puted as the root sum square of the single deviations at the two wavelengths, which can be expressed as

$$\Delta_{tot} = \sqrt{\Delta_{355}^2 + \Delta_{532}^2 + \Delta_{1064}^2}. \tag{7}$$

## 5 Results

### 5.1 Case study on 13 September 2012

During the first ascending spiral, in situ sensors on board the ATR42 were operated in the altitude region from 650 to 5700 m above sea level (hereafter in the paper all altitudes are intended above sea level), covering the 40 min time interval between 19:55 and 20:35 UTC. BASIL was operated in the time interval 19:00–23:00 UTC. Figure 3 illustrates the temporal evolution of the particle backscattering coefficient at 532 nm over the time interval 19:30–21:30 UTC. The figure is illustrated as a succession of 5 min vertical profiles with a vertical resolution of 7.5 m. The figure reveals the presence of a shallow nocturnal boundary layer, which is testified by the presence of an aerosol layer extending up to 500–600 m and the presence of a residual layer extending up to 1500–2100 m.

Wind direction measurements performed by the on-board flight sensors reveal a primarily northerly wind, with direction varying in the range ±30° depending on altitude. The NOAA HYSPLIT Lagrangian back-trajectory model (Draxler and Rolph, 1998; Rolph et al., 2017; Stein et al., 2015) has been used to determine the origin of the sounded air masses. The HYSPLIT model computes air parcel trajectories, CEI but it can also be used to simulate complex transport, dispersion, chemical transformation and deposition mechanisms. A common application of the HYSPLIT model is the back- and forward-trajectory analysis, which is used to determine the origin or destination of the investigated air masses and establish source–receptor relationships.

In the present effort the HYSPLIT model is used to determine air masses trajectories at specific altitude levels in the days preceding their arrival on the lidar site in Candillargues. Specifically, Fig. 4 illustrates back trajectories of the air masses overpassing the lidar site at 20:00 UTC on 13 September 2012 at an altitude of 600 (red line), 4000 (blue line) and 6000 m (green line). The trajectories extend back in time for 5 days, thus illustrating the air masses' path since 20:00 UTC on 8 September 2012.

Air masses reaching the measurement site at altitudes of 600 and 4000 m originated in the vicinity of Iceland and Greenland and passed at low altitudes (< 400 m) over the North Atlantic Ocean and over industrialized areas in France, while air masses at 5826 m originated in the North Atlantic Ocean in the proximity of the Canadian coasts and persisted in a marine environment for almost 5 days before reaching France.

Figure 5 compares the vertical profiles of the measured and simulated particle backscattering coefficient at 355 nm. The measured profile is obtained from the Raman lidar data integrated over the 40 min time interval coincident with the airplane ascent time (19:55–20:35 UTC on 13 September 2012), with a vertical resolution of 300 m. Simulated particle backscatter profiles include all five aerosol components specified above, i.e. the continental–urban component (red dashed line), the continental (rural) component (green dashed line), the Arctic summer component (black dashed line) and

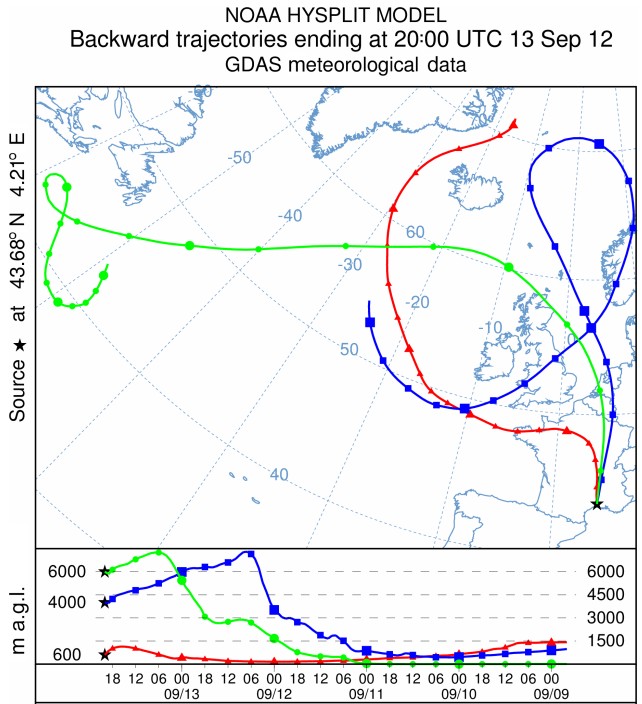

**Figure 4.** Air mass back trajectories at 600 (red), 4000 (blue) and 6000 m (green) ending over the lidar site at 20:00 UTC on 13 September 2012.

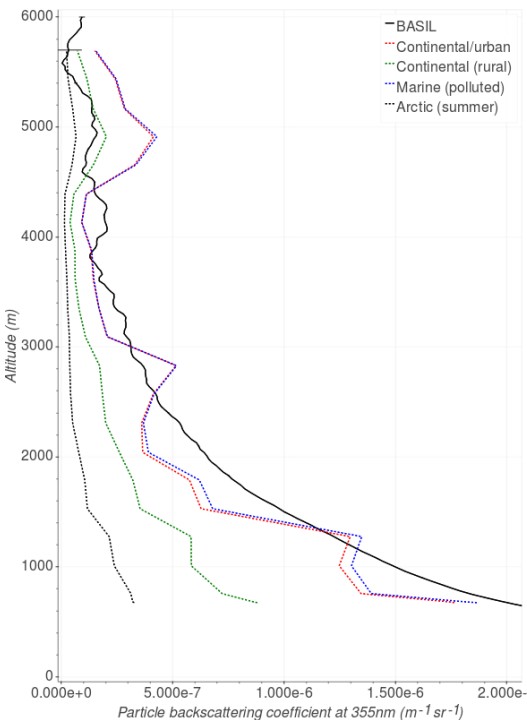

**Figure 5.** Vertical profiles of the measured (black line) and simulated particle backscattering coefficient at 355 nm over the time interval 19:55–20:35 UTC on 13 September 2012. The error bar in lidar measurements accounts for the statistical uncertainty.

the marine (polluted) component (blue dashed line). Figure 5 reveals a good agreement between the measured backscattering coefficient profile at 355 nm and coefficients simulated at this same wavelength assuming a continental–urban aerosol component and a marine (polluted) aerosol component.

The same analysis approach was also applied to the data at 532 nm. Figure 6 compares the vertical profiles of the measured (black line) and simulated (red line) particle backscattering coefficient at 532 nm over the same 40 min time interval on 13 September 2012, again with a vertical resolution of 300 m. Simulated particle backscatter profiles include the five above specified aerosol components. Lidar data at 532 nm are affected by a larger statistical uncertainty than those at 355 nm. Also in this case, the agreement between measured and simulated profiles appears to be quite good up to 3500–4000 m.

Figure 6 reveals that the measured particle backscattering coefficient profile at 532 nm is well reproduced by the simulated profiles at this same wavelength, especially the profiles considering a continental–urban aerosol component and a marine (polluted) aerosol component, with simulated profiles slightly overestimating the measured profile but being within or slightly exceeding the measurement error bar. Deviations between measured and simulated profiles are larger within the aerosol layer centred at 2800 m.

Figure 7 compares the vertical profiles of the measured and simulated particle backscattering coefficient at 1064 nm

over the same 40 min time interval considered in Figs. 5 and 6, again with a vertical resolution of 300 m. Particle backscatter measurements at 1064 nm are affected by a statistical uncertainty larger than the one affecting the measurements at 532 nm. This larger uncertainty is the result of the use of a reduced laser emission power at 1064 nm because of the restrictions imposed by the air traffic control authorities. In this case, the agreement between measured and simulated profiles is poorer but still acceptable up to 2500 m.

Figure 8 illustrates the deviations between the measured and the simulated particle backscattering coefficient profile at 355 nm. The smallest deviations between the two profiles up to 4500 m are obtained when considering the presence of a marine polluted aerosol component (smaller than 53 %, with a mean deviation of 23.2 %). Simulated profiles obtained considering a continental–urban aerosol component (not exceeding 54 %, with a mean deviation of 24.9 %) deviate less only within the altitude interval 1200–1300 m, while deviations are very similar above 2600 m. The simulated profile obtained considering the presence of either a continental rural or an Arctic summer aerosol component largely deviates from the measured profile (up to 80 % and 92 %, respectively, with a mean deviation of 50.9 % and 25.9 %). The Arctic component deviates less only above 4500 m, where the high signal noise level and the limited particle loading make aerosol type discrimination difficult to accomplish.

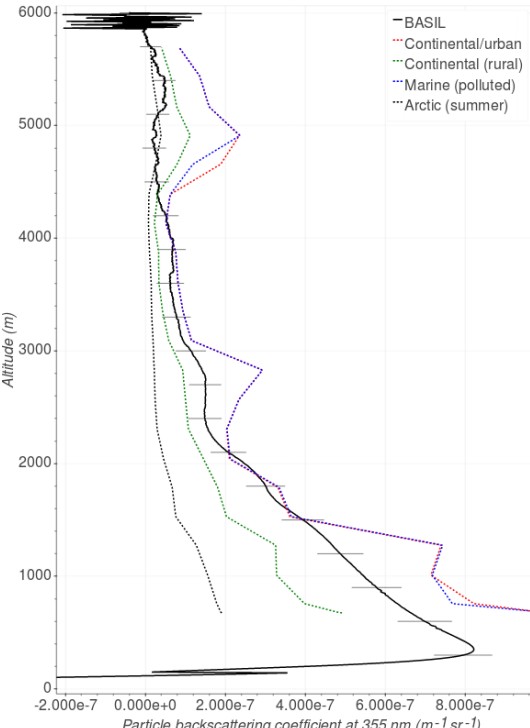

**Figure 6.** Same as Fig. 5 but for the particle backscattering coefficient at 532 nm.

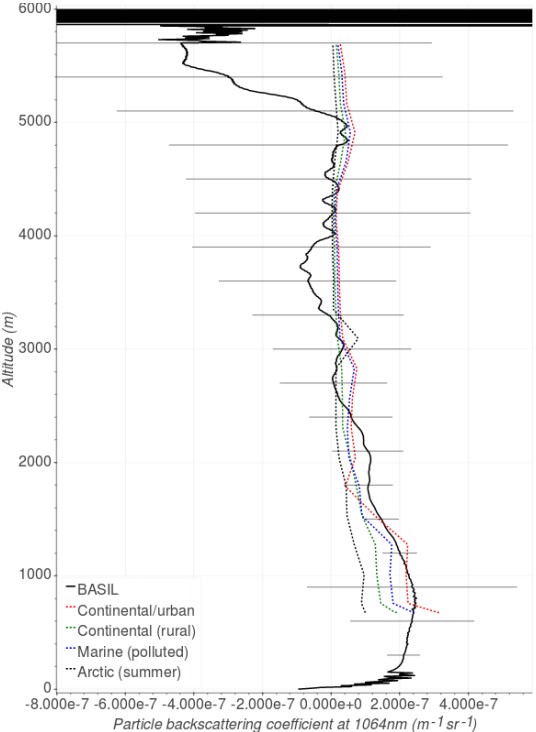

**Figure 7.** Same as Fig. 5 but for the particle backscattering coefficient at 1064 nm.

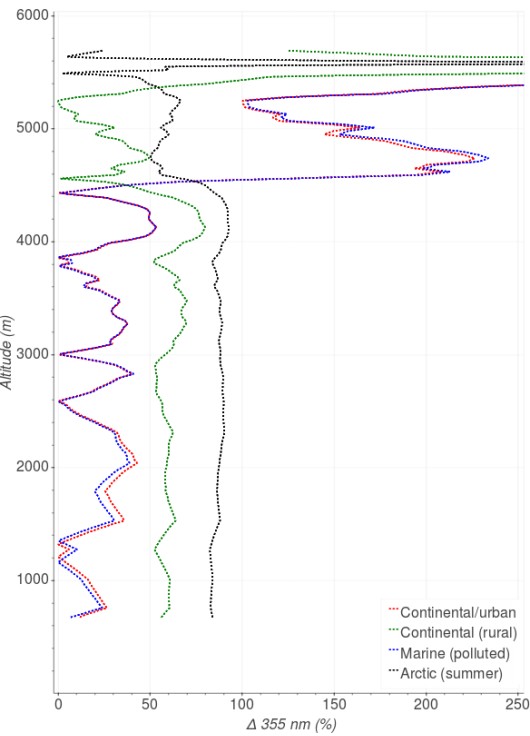

**Figure 8.** Deviation, expressed in percentage, between measured and simulated particle backscattering coefficient profiles at 355 nm. Simulated profiles are Arctic summer (black dashed line), continental–urban (red dashed line), marine (polluted) (blue dashed line) and continental (rural) (green dashed line).

Figure 9 illustrates the deviations between the measured and the simulated particle backscattering coefficient profile at 532 nm. Again, the maximum altitude for aerosol type retrieval is 4340 m. The smallest deviations between measured and simulated particle backscattering coefficient profiles are obtained when considering the presence of a continental–urban aerosol component (not exceeding 105 %, with a mean value of 30.8 %) or a marine polluted aerosol component (smaller than 106 %, with a mean value of 30.9 %), while simulated profiles obtained considering the presence of either a continental rural or an Arctic summer aerosol component largely deviate from the measured profile (up to 60.6 % and 87 %, respectively, with a mean deviation of 39.6 % and 79.2 %). The only exception is given by the interval 2300–3000 m, where the simulated profile obtained considering a rural aerosol component deviates less.

Figure 10 illustrates the deviations between the measured and the simulated particle backscattering coefficient profile at 1064 nm considering altitudes up to 2500 m. The smallest deviations between the two profiles over the considered altitude range are obtained when considering the presence of a continental–urban aerosol component (not exceeding 61.4 %, with a mean deviation of 21.2 %). Deviations between measured and simulated profile obtained considering a marine

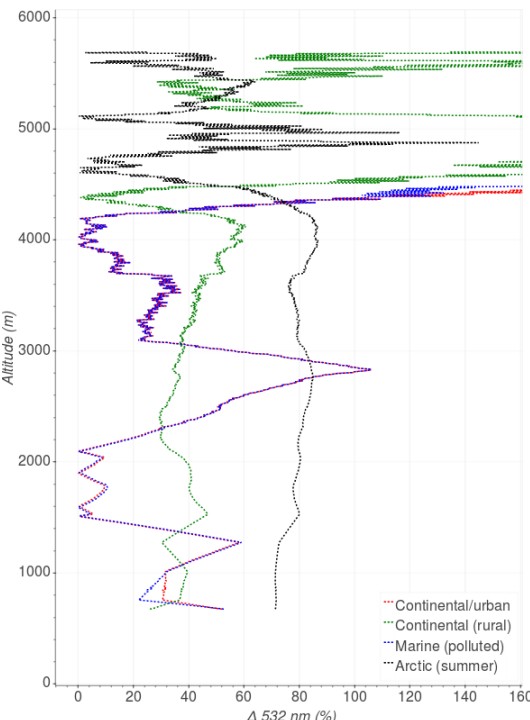

**Figure 9.** Same as Fig. 8 but obtained considering particle backscattering coefficient profiles at 532 nm.

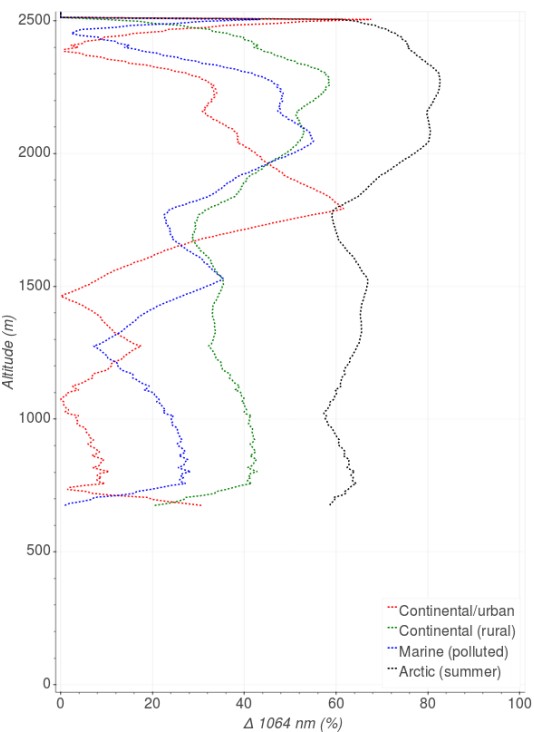

**Figure 10.** Same as Fig. 8 but obtained considering particle backscattering coefficient profiles at 1064 nm up to 2500 m.

polluted aerosol component are slightly larger (smaller than 55 %, with a mean deviation of 28.6 %), while the simulated profile obtained considering the presence of either a continental rural or an Arctic summer aerosol component largely deviate from the measured profile (up to 58 % and 82.7 %, with a mean deviation of 40.9 % and 67.3 %, respectively). Again, the only exception is found in the interval 1600–1900 m, where the simulated profile obtained considering the marine polluted aerosol component deviate less.

The overall deviation was calculated for the five distinct aerosol components. Figure 11 illustrates the overall deviations between the measured and the simulated particle backscattering coefficient profiles at 355, 532 and 1064 nm for the different aerosol components. In order to facilitate the interpretation of results, the overall deviation between measured and simulated particle backscattering coefficient profiles, for the different aerosol components, has been plotted together with the measured particle backscattering profiles at all wavelengths (Fig. 12). In the lowest portion of the atmosphere up to 1700 m, i.e. inside the planetary boundary layer, the continental–urban aerosol component is predominant. The upper layer between 1700 and 2400 m is characterized by the presence of a maritime aerosol component in the lower part and again an urban aerosol component in the upper part. Deviations including the particle backscattering coefficient at 1064 nm were computed up to 2500 m because of the high statistical noise of the 1064 nm lidar signal. Additional

layers are visible in the altitude range 2400–3100 and 3800–4500 m. Above 2400 m simulations based on the urban and maritime components show similar deviations from measurements, except in the central part of layer between 2600 and 2900 m and between 4300 and 4500 m, where rural aerosols deviate less. HYSPLIT back-trajectory analysis confirms that the sounded air masses in the previous days overpass industrialized areas in France, Belgium and England.

## 5.2 Case study on 2 October 2012

A second flight took place on 2 October 2012. During the ascending path, in situ sensors on board the ATR42 were operated in the altitude region from 680 to 5700 m, covering a 44 min time interval between 19:43 and 20:27 UTC. BASIL was operated over the time interval 16:00–24:00 UTC.

Wind direction measurements performed by the on-board flight sensors reveal a north-westerly wind, with direction varying in the range 220–320° depending on altitude. Figure 13 shows the 5-day back trajectories from the NOAA HYSPLIT model at 600, 4000 and 6000 m (in red, blue and green, respectively), ending on the lidar site at 20:00 UTC on 2 October 2012.

Back-trajectory analysis results reveal that air masses reaching the measurement site at an altitude of 600 m originated in the North Atlantic Ocean, south of Iceland, and have passed at low altitudes (500–600 m) over highly anthropogenic continental areas (Ireland, England and northern

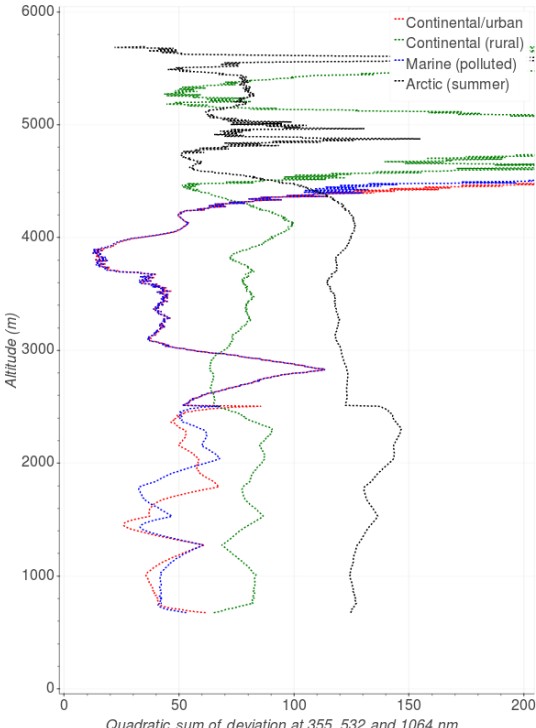

**Figure 11.** Total deviation, in percentage, between measured and simulated particle backscattering coefficient profiles at 355, 532 and 1064 nm (up to 2500 m) for the different aerosol components. Simulated profiles are Arctic summer (black dashed line), continental–urban (red dashed line), marine (polluted) (blue dashed line) and continental (rural) (green dashed line).

France). A different path characterizes air masses at 4000 m. These originated over the North Atlantic Ocean, offshore of the Canadian coast, and overpassed an area north of the Azores over the northern coast of Spain before reaching the measurement site. Finally, air masses reaching the measurement site at 6000 m which originated over the North Pacific Ocean overpassed Canada, the North Atlantic Ocean, the northern coast of Spain and finally reached the measurement site.

In the analysis of this second case study, we applied the same methodology considered for the first case study (1991). As for the previous case study, given the microphysical parameters and aerosol typology for each of the three given modes, the number of particles has been varied in order for the theoretical distribution to match the experimental distribution computed with the OPC data, with the matching between the experimental and theoretical distributions again obtained through a best fit procedure. The modal radius, standard deviation and refractive index reported by d'Almeida et al. (1991) for the different considered aerosol components are listed in Table 1.

Figure 14 illustrates the vertical profiles of measured (black line) and simulated particle backscattering coefficient

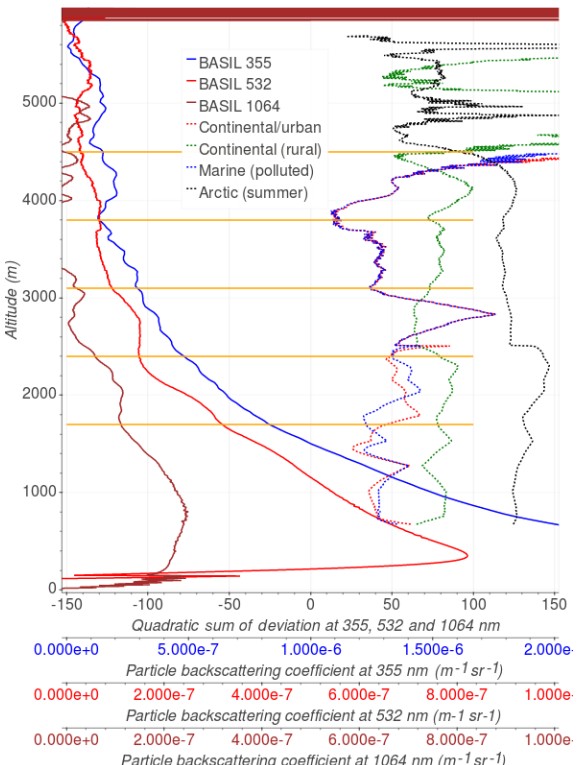

**Figure 12.** Total deviation, in percentage, between measured and simulated particle backscattering coefficient profiles for the different aerosol components (Arctic summer: black dashed line, continental–urban: red dashed line, marine polluted: blue dashed line; continental rural: green dashed line) and measured particle backscattering profiles at both 355 (blue line) and 532 nm (red line). The horizontal blue and red axes refer to the particle backscattering coefficient at 355 and 532 nm, respectively, while the horizontal black axis refers to the total deviations. Horizontal orange lines are also drawn at specific altitudes to identify different aerosol types in support of the interpretation of the reported results.

at 355 nm over the 44 min time interval between 19:43 and 20:27 UTC on 2 October 2012. Simulated particle backscatter profiles include all five aerosol components specified above, i.e. the continental–urban component (red dashed line), the continental (rural) component (green dashed line), the Arctic summer component (black dashed line) and the marine (polluted) component (blue dashed line). Figure 14 reveals a good agreement between the measured backscattering coefficient profile at 355 nm and coefficients simulated at this same wavelength assuming a continental–urban aerosol component and a marine (polluted) aerosol component.

We also applied this same analysis approach to the data at 532 nm, with Fig. 15 illustrating the vertical profiles of the measured and simulated particle backscattering coefficient at 532 nm over the same time interval considered in Fig. 14. Again, simulated particle backscatter profiles include the five above-specified aerosol components. Figure 15 reveals that the measured particle backscattering coefficient

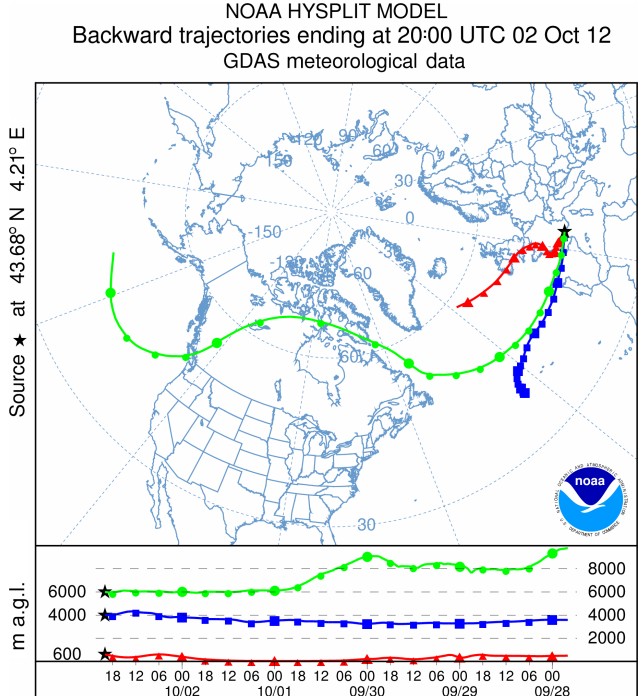

**Figure 13.** Back trajectories at 600 (red), 4000 (blue) and 6000 m (green) ending on the lidar site at 20:00 UTC on 2 October 2012.

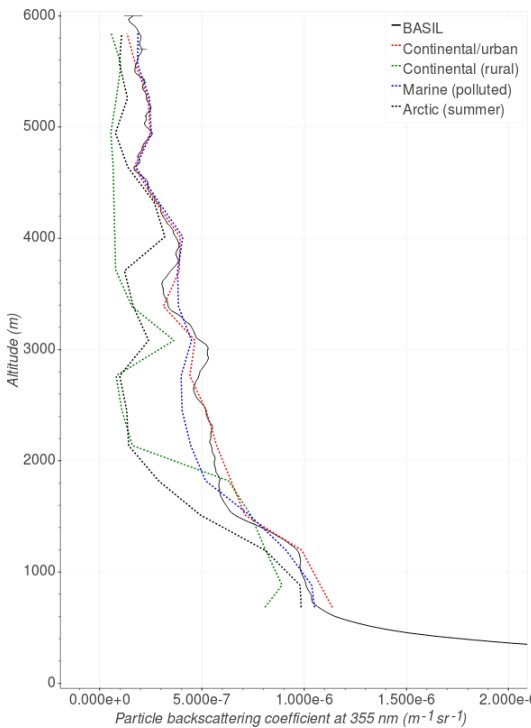

**Figure 14.** Vertical profiles of measured (black line) and simulated particle backscattering coefficient at 355 nm over the time interval 19:43–20:27 UTC on 2 October 2012. Simulated particle backscatter profiles include five distinct components: continental–urban (red dashed line), continental (rural) (green dashed line), Arctic summer (black dashed line) and marine (polluted) (blue dashed line). The error bar in lidar measurements accounts for the statistical uncertainty.

profile at 532 nm is well reproduced by the simulated profiles at this same wavelength, especially the profiles considering a continental–urban aerosol component and a marine (polluted) aerosol component, with simulated profiles slightly underestimating the measured profile but being within or slightly exceeding the measurement error bar. Deviations between measured and simulated profiles are larger within the aerosol layers centred at 3000 and 4000 m. Due to the limited laser power at 1064 nm for this specific measurement session, measured profiles of the particle backscattering coefficient at 1064 nm are characterized by high statistical noise, which prevents us from considering the use of the comparison between measured and simulated particle backscatter profiles at this wavelength in the present analysis.

Figure 16 illustrates the deviations between the measured and the simulated particle backscattering coefficient profiles at 355 nm. The smallest deviations between the measured and the simulated particle backscattering coefficient profile over the considered altitude range are obtained when considering the presence of a continental–urban aerosol component (not exceeding 15 % up to 5000 m, with a mean deviation of 5.9 %). Deviations between the measured and simulated profile obtained considering a marine polluted aerosol component slightly exceed these values (smaller than 20 % up to 5000 m, with a mean deviation of 9.5 %), while the simulated profile obtained considering the presence of either a continental rural or an Arctic summer aerosol component largely

deviates from the measured profile (up to 80 %, with a mean deviation of 50.9 % and 25.9 %, respectively).

Figure 17 illustrates the deviations between measured and simulated particle backscattering coefficient profiles at 532 nm. Again, the smallest deviations between the two profiles over the considered altitude range are obtained when considering a continental–urban aerosol component (not exceeding 50 % up to 5000 m, with a mean deviation of 25.9 %), with the only exception for the interval 3100–3700 m, where the simulated profile obtained considering a marine polluted aerosol component deviates less. Above 3700 m simulated profiles obtained considering a continental–urban and a marine polluted aerosol component equally deviate from the measured profile.

In the attempt to simultaneously minimize deviations between measured and simulated particle backscattering profiles at both 355 and 532 nm, following Eq. (7), a total deviation can be computed as the root sum square of the single deviations at the two wavelengths, which can be expressed as

$$\Delta_{\mathrm{tot}} = \sqrt{\Delta_{355}^2 + \Delta_{532}^2}. \tag{8}$$

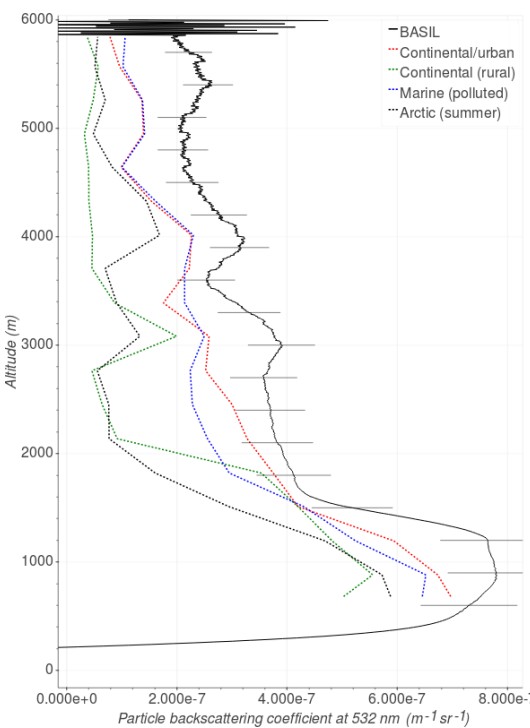

**Figure 15.** Same as Fig. 14 but with particle backscattering coefficient profiles at 532 nm.

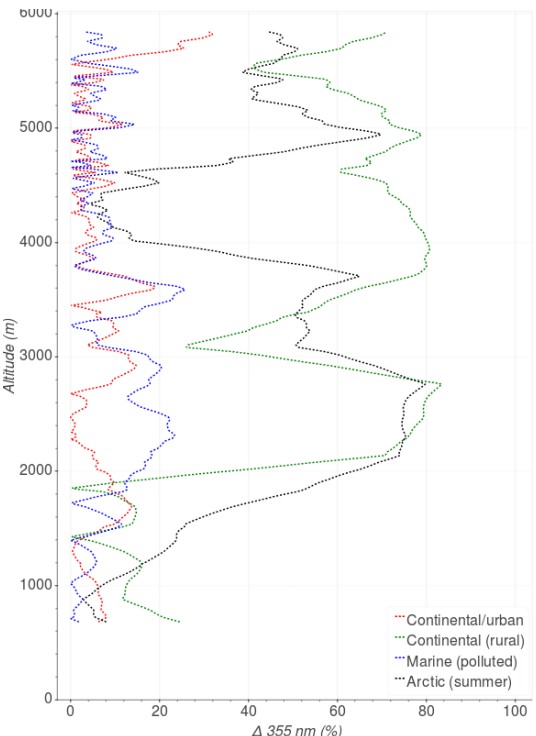

**Figure 16.** Deviation, expressed in percentage, between measured and simulated particle backscattering coefficient profiles at 355 nm. Simulated profiles are Arctic summer (black dashed line), continental–urban (red dashed line), marine (polluted) (blue dashed line) and continental (rural) (green dashed line).

This quantity was calculated for the five distinct aerosol components. Figure 18 illustrates the total deviations between the measured and the simulated particle backscattering coefficient profiles at 355 and 532 nm for the different aerosol components. In order to facilitate the interpretation of these results, the total deviation between measured and simulated particle backscattering coefficient profiles for the different aerosol components has been plotted together with the measured particle backscattering profiles at both 355 and 532 nm (Fig. 19).

Figure 19 allows the following considerations to be made. In the lowest portion of the atmosphere, up to an altitude of ∼ 1300 m (altitude 1), aerosol particles are most likely characterized by a predominant continental–urban component. This aerosol layer extends up to ∼ 1600 m, which is the altitude at which the boundary layer height is located, as also indicated by the simultaneous radiosonde data (not shown here). In the upper portion of the boundary layer, in the vertical interval 1300–1600 m, deviations associated with continental–urban, marine polluted and continental rural components overlap, which suggests CE2 that all three aerosol components are possible. However, while this upper portion of the boundary layer is typically characterized by entrainment effects (interfacial region), which may allow different aerosol components to be ingested, the continental–urban component is likely to be the predominant component.

Above the top of the boundary layer and up to ∼ 2700 m (altitude 2), particle backscatter decreases with altitude. The typology analysis suggests continental–urban aerosols likely to be the predominant component, as in fact total deviation between the measured and the simulated particle backscattering coefficient profile for this aerosol component is far lower than for all other aerosol components.

In the altitude interval 2700–3600 m (altitudes 2–3, with max. at 3000 m) the measured particle backscatter profiles reveal the presence of a distinct aerosol layer. The typology analysis indicates that both the continental–urban and the marine polluted components are possible. An additional distinct aerosol layer is found in the altitude interval 3600–4600 m (altitudes 3–4, with max. at 4000 m). Again, the typology analysis suggests the continental–urban component is possible. Sounded aerosol particles at 3000 and 4000 m are compatible with continental polluted aerosols, this possibility being supported by the back-trajectory analysis at 3000 and 4000 m.

A sensitivity study has also been carried out to assess the variability of the results to changes of specific size and microphysical parameters' values. The sensitivity study reveals that the considered methodology for aerosol typing is successfully applicable in the altitude region up to 3900 m, as

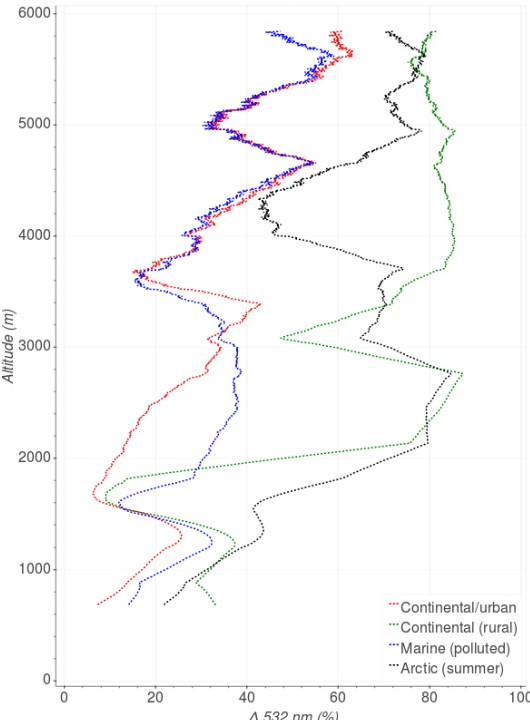

**Figure 17.** Same as Fig. 16 but obtained considering particle backscattering coefficient profiles at 532 nm.

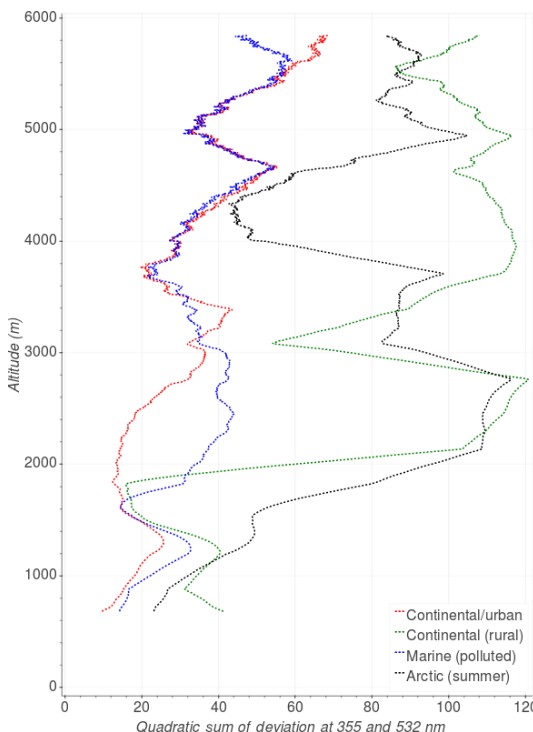

**Figure 18.** Total deviation, in percentage, between measured and simulated particle backscattering coefficient profiles at 355 and 532 nm for the different aerosol components. Simulated profiles are Arctic summer (black dashed line), continental–urban (red dashed line), marine (polluted) (blue dashed line) and continental (rural) (green dashed line).

in fact above this altitude the statistical uncertainty affecting the lidar signals is high, and this severely reduces the effectiveness of the aerosol typing methodology. The sensitivity analysis also reveals that in the lower levels, typically within the boundary layer where aerosol loading is larger, deviations between measured and simulated particle backscattering coefficient at the three wavelengths may vary by up to 20 % as a result of a ±5 % variability of specific size and microphysical parameters (for example, the real part of the refractive index), which certainly reduces confidence in the aerosol typing approach but is not compromising its outcome. Based on the results from this study we may conclude that the use of particle backscattering measurements at two wavelengths in combination with OPC measurements allows CE3 a sufficiently reliable assessment of the aerosol types to be obtained, which can be verified and refined based on the use of back-trajectory analyses.

## 6 Summary

During HyMeX-SOP1, the Raman lidar system BASIL was deployed in Candillargues (southern France) and operated almost continuously over a 2-month period in the time frame October–November 2012. Dedicated flights of the French research aircraft ATR42 were carried out in the framework of the EUFAR-WaLiTemp Project. The ATR42 payload included in situ sensors for turbulence measurements, as well

as aerosol and cloud microphysics probes, together with an optical particle counter (GRIMM Aerosol Technik GmbH, model: Sky-OPC 1.129) capable of measuring particle number concentration in the size interval 0.25–2.5 μm. A specific flight pattern was considered for the purpose of this study, with the aircraft making spirals up and down around a central location approximately 20 km eastward of the lidar site. Vertical profiles of the particle backscattering coefficient at 355, 532 and 1064 nm have been simulated through the use of a Mie scattering code, using the data provided by the optical particle counter. The simulated particle backscatter profiles have been compared with the profiles measured by the lidar Raman system BASIL. Results from two selected case studies (on 13 September and on 2 October 2012) are reported and discussed. An analysis approach based on the application of a Mie scattering code is considered and applied. The approach ultimately allows the sounded aerosol types to be inferred. The added value of the reported methodology is represented by the possibility to infer the presence of different aerosol types based on the use of multi-wavelength Raman lidar measurement from a ground-based system in combination with an independent measurement of the particle concentration profile (in our case we are using the one coming from an optical particle counter mounted on board an air-

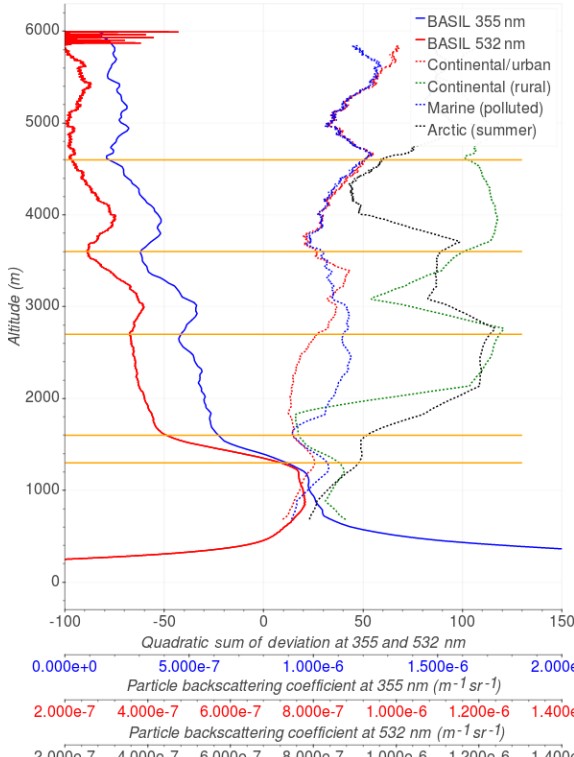

**Figure 19.** Total deviation, in percentage, between measured and simulated particle backscattering coefficient profiles for the different aerosol components (Arctic summer: black dashed line, continental–urban: red dashed line, marine polluted: blue dashed line; continental rural: green dashed line) and measured particle backscattering profiles at both 355 (blue line) and 532 nm (red line). The horizontal blue and red axes refer to the particle backscattering coefficient at 355 and 532 nm, respectively, while the horizontal black axis refers to the total deviations. Horizontal orange lines are also drawn at specific altitudes to identify different aerosol types in support of the interpretation of the reported results.

craft overpassing the lidar site). This methodology is applicable when sounded particles are spherical or almost spherical, which allows for the Mie scattering theory to be applied for the determination of the particle backscattering coefficient. The HYSPLIT-NOAA back-trajectory model was used to verify the origin of the sounded aerosol particles.

Five different aerosol typologies are considered, i.e. continental polluted, clean continental–rural, urban, maritime polluted and clean–polar, with their size and microphysical properties taken from literature. The approach leads to an assessment of the predominant aerosol component based on the application of a minimization approach applied to the deviations between measured and simulated particle backscattering profiles at 355 and 532 nm and for the first test case study also at 1064 nm, considering all five aerosol typologies.

The application of this approach to the case study on 13 September 2012 suggests the presence of urban and maritime aerosols throughout the entire vertical extent of

sounded column, except in the altitude region 2600–2900 and 4300–4500 m ranges, where the presence of a rural component is likely to be possible. The application of the approach to the case study on 2 October 2012 reveals that continental–urban aerosols are likely to be the predominant components up to ∼ 1600 m, while the two distinct aerosol layers located in the altitude regions 2700–3600 (with max. at 3000 m) and 3600–4600 m (with max. at 4000 m) are identified to likely consist of continental–urban and/or marine polluted aerosols, respectively. The correctness of the results has been verified based on the application of the HYSPLIT-NOAA back-trajectory model, with the analysis extend backing in time for 5 days allowing the origin of the sounded aerosol particles to be assessed.

Finally, a sensitivity study has been carried out to assess the variability of the aerosol typing approach to varying size and microphysical parameters. The study reveals that the reported approach is successfully applicable in the altitude region up to ∼ 4 km, while above this altitude the sensitivity of the approach is substantially reduced by the high statistical uncertainty affecting lidar signals. The sensitivity study also reveals that the within-boundary-layer deviations between measured and simulated particle backscattering coefficients at 355, 532 and 1064 nm may vary up to 20 % as a result of ±5 % variability of specific size and microphysical parameters. Such results reveal that the application of the reported approach, based on the use of particle backscattering measurements at two wavelengths in combination with OPC measurements, allows a sufficiently reliable assessment of aerosol typing to be obtained.

*Data availability.* Data used in this study, together with the related metadata, are available from the public data repository HyMeX database, which is freely accessible by all users through the following link: http://mistrals.sedoo.fr/HyMeX/ TS2 .

*Author contributions.* PDG designed and developed the main experiment, and MC designed and developed the additional receiving unit. PDG, MC, DS, AS and DS carried out the measurements. DS, AS and DS developed the data analysis algorithms and carried out the data analysis. DS and PDG prepared the manuscript with contributions from DS.

*Competing interests.* The authors declare that they have no conflict of interest.

*Special issue statement.* This article is part of the special issue "Hydrological cycle in the Mediterranean (ACP/AMT/GMD/HESS/NHESS/OS inter-journal SI)". It is not associated with a conference.

*Acknowledgements.* This work is a contribution to the HyMeX Program supported by MISTRALS and ANR IODA-MED grant ANR-11-BS56-0005. This research effort was supported by the European Commission under the European Facility for Airborne Research of the Seventh Framework Programme (WaLiTemp project). This research effort was also supported by the project "Smart Cities – Basilicata" by the Italian Ministry of Education, University and Research. The authors gratefully acknowledge NOAA Air Resources Laboratory (ARL) for the provision of the HYSPLIT transport and dispersion model used in this publication.

*Review statement.* This paper was edited by Domenico Cimini and reviewed by three anonymous referees.

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

**Remarks from the language copy-editor**

CE1     Please note that the comma was inserted as requested, but it was necessary to add "it". The clause would otherwise not be grammatically correct.

CE2     Please note that the requested change could not be implemented because it was grammatically incorrect. This is the correct construction to be used with the verb "suggest".

CE3     Please note that the requested change could not be made because it was grammatically incorrect. This is the correct construction to be used with the verb "allows".

**Remarks from the typesetter**

TS1     Please note that such changes cannot be inserted at this stage. If you insist on these corrections we have to ask for the editor's approval.

TS2     Please provide a reference list entry including authors/creators, title, and date of last access.