# Peer review of "Characterization of atmospheric aerosol optical properties based on the combined use of a ground-based Raman lidar and an airborne optical particle counter in the framework of the Hydrological Cycle in the Mediterranean Experiment – Special Observation Period 1"

_Atmospheric Measurement Techniques, 2018_

## Referee Comment (RC1) · Anonymous Referee #2 · 21 Nov 2018

The manuscript fits within the journal scope, as assessing the optical and microphysical aerosol properties is crucial to reducing the uncertainty in climate model temperature rise (for example).

Nevertheless, the proposed methodology is questionable, because the profile for aerosol backscattering coefficient is not directly measured, but retrieved under as-

sumptions. The result is then compared with the aerosol backscattering profile obtained combining aircraft measurements of aerosol concentration and Mie scattering model. The simulated profiles come too many assumptions (i.e., the aerosol type, size distribution and so on). It doesn't make sufficient sense. I would instead suggest to retrieve the aerosol particle concentration using the three lidar wavelengths (as showed in Veselovskii et al.) and compare with OPC direct measurements.

In line 141 is stated that the air masses over the measurement site at higher altitude contain marine aerosol. This kind of analysis is way too simple, as the presence of upper air marine salt is rather sporadic and under very specific characteristics (e.g. hurricanes). Upper atmosphere marine aerosols were detected occasionally and just for very specific meteorological conditions (e.g. hurricanes, Sassen, 2003)

Summary is pretty confusing. Line 330: the two approaches are not slightly different, but very different! Then the two approaches don't allow to infer the aerosol types. For the first case study , the aerosol species are assumed from Hysplit back-trajectories analysis. The sentence on line 333 is immediately contradicted by line 334.

As a general comment, it is missing a discussion on why the authors investigate those two different approaches and which are the implications based on theanalysis. Currently it looks like an exercise in style.

Major revisions are needed before publication, but I am confident that the authors will succesfully address the previously raised issues.

Specific comments:

Line 36: "smaller" is not the right word . Please rather use "shorter"

Line 59: please consider using " from.. to. . ."

Line 122: how the particle backscattering coefficient is retrieved ? Raman channel? Please specify.

Line 151: In the equation is missing a bracket (please use square brackets). Moreover, some terms are poorly or not at all described.

Lines 156-158: the sentence is not clear at all, please rephrase.

Line 186, again, smaller is not a good choice. Use rather "poorer"

Fig. 16 X-axis label is missing

---

## Referee Comment (RC2) · Anonymous Referee #3 · 24 Dec 2018

This paper deals with the combined analysis of an optical particle counter deployed on airplane and lidar with the objective of providing aerosol typing at different altitudes. The topic is sounds and deserve a publication. The paper is also generally well-written and structured, although eventually lack of consistent scientific conclusions about the method proposed. Before recommending the manuscript publication I have several

issues that needs to be addressed.

MAJOR CONCERNS

My main concern is about the conclusions obtained in both studies cases when aerosol loads is very low, which is observed in the free troposphere for your study cases. Authors should address this limitation of their methodology by for example doing sensitivity tests using synthetic data. Such analysis will help to better understand the differences you obtain in your analyses.

It is necessary to improve the methodology section about how you obtain different aerosol types from the fitting procedure you propose. It is not clear to me what aerosol parameters you are obtaining. I would recommend adding references to Table 1.

It is not clear to me the capabilities of BASIL and what you are actually using. You say that they are capable of obtaining independent extinction and backscattering at 355 and 532 nm by Raman technique. But at some points in the manuscript you mention that you use only elastic signals at 532 nm. If this is true, it means that you are using the classical Klett method for obtaining backscattering method that imply constant lidar ratios and thus homogenous aerosol in the entire column. What is then the sense of applying your methodology for aerosol typing? This question is also applicable for backscttering at 1064 nm.

The two study cases are not very different. The area is usually affected by Saharan dust transport. The paper will definitly gain more interest if an study case affected by dust transport is included.

MINOR CONCERNS

Why using HYSPLIT at different altitues between the two examples? Please unify criterion. There should not be dramatical changes in your interpretation.

I agree with the previous referee about the interpretation of marine aerosol at high levels for the case study on 13th September 2012. See my previous comments about

the applicability of your methodology for very low aerosol loads.

Lines 30-35: Please provide references.

Line 164: 'particles at higher altitudes are internal mixtures of...' It is impossible to know if they are internal or external mixtures with the instrumentation and methodology you used. Please correct.

Figure 1 needs improvements as in its current form does not provides any information. Why not include flight tracks?

What are the effects of incomplete overlap in your methodology?

Line 325-326. 'decreasing number concentration of the same aerosol type (Continental/Urban) as a result of the progressive attenuation of the underlying convective activity' I do not understand this sentence. Please clarify. Conclusions must be improved. What is the added value of your methodology? Under which circunstances is applicable? Which aerosol properties are you providing?
* * *

---

## Author Comment (AC1) · 18 Feb 2019

Responses to both referees are included in the attached supplement docuement.

Please also note the supplement to this comment:

https://www.atmos-meas-tech-discuss.net/amt-2018-268/amt-2018-268-AC1-supplement.pdf

---

## Author Comment (AC2) · 18 Feb 2019

Dear Editor,

We are very grateful to the two referees for their appropriate and constructive suggestions and for their proposed corrections to improve the paper. We have addressed all issues raised and have modified the paper accordingly. If you and the referee agree on that, we are also ready to submit a revised version of the paper where all these changes have been introduced. We believe that, thanks to their precious inputs, the manuscript has now sensitively improved. Below is a summary of the changes we made and our specific responses to the referees' comments and recommendations.

**Summary of the changes**

**(in black is the original comments of the referee and in red our responses)**

**Anonymous Referee #2**

The manuscript fits within the journal scope, as assessing the optical and microphysical aerosol properties is crucial to reducing the uncertainty in climate model temperature rise (for example). Nevertheless, the proposed methodology is questionable, because the profile for aerosol backscattering coefficient **is not directly measured**, **but retrieved under assumptions**. The result is then compared with the aerosol backscattering profile obtained combining aircraft measurements of aerosol concentration and Mie scattering model. The simulated profiles come too many assumptions (i.e., the aerosol type, size distribution and so on). It doesn't make sufficient sense.

The authors are not sure they properly understand this comment from the referee as in fact the purpose of the present paper, as stated from the first sentence of the Abstract, is to compare: "…vertical profiles of particle backscattering coefficient at 355, 532 and 1064 nm **measured by the lidar Raman** system BASIL … with **simulated particle backscatter profiles obtained through a Mie scattering code** based on the use of simultaneous and almost **co-located profiles of optical and microphysical aerosol properties** measured by an air-borne optical particle counter. So, differently from what stated above by the referee, the aerosol backscattering coefficient profile **is directly measured by the Raman lidar** and compared simulated profiles.

We agree that simulated profiles implies a certain number of assumptions (primarily the aerosol type and size distribution), but a certain number of constraints can be applied to the assumed parameters based on the additional information available, besides the Raman lidar and the air-borne in-situ data. In this respect, it is to be specified that an assessment of the aerosol type and the different possible components in it is supported and verified based on the use of back-trajectory analyses, which allow to identify aerosol origin and its specific path. This is true for both proposed approaches. The possibility to verify aerosol typing through back-trajectory analyses has been considered by a variety of authors in the past (among others, Man and Shih, 2001; Methven et al., 2001; Estellés et al., 2007; Toledano et al., 2009). In this regard, the following sentence was introduced in the text: "Back-trajectory analyses from a Lagrangian model ((HYSPLIT)) are used in support of the assessment of aerosol types (Man and Shih, 2001; Methven et al., 2001; Estellés et al., 2007; Toledano et al., 2009)." Nevertheless, we tried to further support and strengthen the considered assumptions based on additional arguments and a specific sensitivity analysis (see specific comments below).

I would instead suggest to retrieve the aerosol particle concentration using the three lidar wavelengths (as showed in Veselovskii et al.) and compare with OPC direct measurements.

We agree that the use of the particle size and microphysical retrieval approach by Veselovskii et al. Authors had a specific interaction with Dr. Igor Veselovskii on this issue during the drafting phase of the paper. However, an appropriate application of this approach imposes the use of particle

backscatter and extinction profiles characterized by very small statistical uncertainties (5-10 %), otherwise the retrieval leads to unstable results. In this respect, it is to be specified that, due to the limited aerosol loading present in the atmosphere for these case studies when simultaneous ground-based Raman lidar and aircraft in-situ measurements were available, the uncertainty affecting the measured particle backscatter and extinction profiles is not sufficiently low to allow an effective application of this approach. This is primarily true for the particle backscatter measurements at 1064 nm. This aspect is now clearly specified in the text, where the following sentences have been introduced: "The possibility to retrieve the particle size and microphysical properties from multi-wavelength measurements of the particle backscattering and extinction coefficient has been demonstrated by several authors (among others, Müller et al., 2001, and Veselovskii et al., 2002) based on the application of a retrieval scheme employing Tikhonov's inversion with regularization, which apply Mie scattering theory to ensemble of particles with spherical shape. However, an appropriate and effective application of this approach imposes the use of particle backscatter and extinction profiles with a statistical uncertainty not exceeding 5-10 %. Multi-wavelength Raman lidar measurements of the particle backscattering and extinction coefficient for the considered case studies were not characterized by such low level of uncertainty, this being especially true for the particle backscatter measurements at 1064 nm."

In line 141 is stated that the air masses over the measurement site at higher altitude contain marine aerosol. This kind of analysis is way too simple, as the presence of upper air marine salt is rather sporadic and under very specific characteristics (e.g. hurricanes). Upper atmosphere marine aerosols were detected occasionally and just for very specific meteorological conditions (e.g. hurricanes, Sassen, 2003).

We agree with the referee that typically the presence of upper air marine salt is rather sporadic and under very specific characteristics (e.g. hurricanes). In this specific case the presence of marine salt at upper levels affected by the occurrence of nearby hurricanes cannot be excluded. In this regard, it is to be specified that during the field campaign a very intense hurricane (hurricane Nadine) took place in the area near Azores, covering a long time period between 10 September and 04 October 2012, with the two selected case studies considered in this paper being on 13 September and 02 October 2012. More specifically, the back-trajectory analysis carried out with a Lagrangian model (HYSPLIT) reveals that the air mass overpassing the lidar site at an altitude of 4000 m on 2 October 2012 was overpassing few days earlier (29-30 September 2012) an area approximately 1000 km North of the Nadine hurricane. However, in the revised version of the paper we removed from the interpretation of the observations the specific reference to marine aerosols in the upper troposphere, aiming to investigate further on this point in a future research effort. Additionally, the hypothesis of observing an internal mixture of continental polluted aerosols with marine salt, i.e. an aerosol coming from a marine polluted environment, has been removed from the analysis and the simulations. In this respect, it is to be also specified that the paper has been heavily reshuffled and now both case studies are analyzed with the same approach previously used for the second case study. **This approach is not using back-trajectory analysis to make preliminary assumption of the aerosol type**, but is using back-trajectory analysis only afterward as a verification tool. The text of the paper was modified accordingly. Consequently, the sentence pointed by the referee in former line 161 (""Based on the involved air masses, we formulate the hypothesis that …") was removed.

Summary is pretty confusing. Line 330: the two approaches are not slightly different, but very different! Then the two approaches don't allow to infer the aerosol types. For the first case study , the aerosol species are assumed from Hysplit back-trajectories analysis.

We agree with the referee that the two considered approaches are quite different. Therefore in the modified version of the paper we decided to use a single approach throughout the paper. For this purpose we selected the second approach, as in fact this approach is not using back-trajectory

analysis to make preliminary assumption of the aerosol type. This approach allows to infer the aerosol types and verify and to verify likelihood of the hypothesis and assumptions considered for aerosol types and size/microphysical properties. For both case studies, back-trajectory analysis carried out with the Lagrangian model HYSPLIT are used to verify aerosol typing and confirm measurement/simulation results. For both case studies, literature values of the refractive index are considered at different altitudes in combination with concentration measurements by the OPC to simulate "synthetic" backscattering profiles through a Mie scattering model; then, particle backscattering coefficient measured by the lidar Raman are compared with the synthetic profiles.

The sentence on line 333 is immediately contradicted by line 334.

The two sentences have been modified in order to remove the contradiction. Now the text reads: "Again, the typology analysis suggests the Continental/Urban component to be possible. Sounded aerosol particles at 3000 and 4000 m are compatible with continental polluted aerosols,  this possibility being confirmed by the back-trajectory analysis at 3000 and 4000 m."

Line 36: "smaller" is not the right word . Please rather use "shorter"

The sentence has been modified and now reads: "Any experiment aimed at characterizing the temporal evolution of aerosol microphysical properties would require several consecutive balloon launches or flights, with the time lag between two consecutive launches/flights unlikely being shorter than 1 hour, with a consequent detriment of the temporal resolution."

Line 59: please consider using " from.. to. . ."

The sentence has been modified and now reads: "The system was deployed in Candillargues (Southern France) in the period from August to November 2012, in the frame of the Hydrological cycle in the Mediterranean Experiment (HyMeX) Special Observation Period 1 (SOP1)."

Line 122: how the particle backscattering coefficient is retrieved ? Raman channel?
Please specify.

The method to determine the particle backscattering coefficient was illustrated in section 5.1 and has now been moved to the Methodology section (section 4), where the following paragrath - slightly different from the previous one - is present: "Measured profiles of the particle backscattering coefficient profiles at 355 and 532 nm are obtained from the Raman lidar signals through the application of the Raman techniques, which relies on the ratio between the 355/532 nm elastic signal and the simultaneous molecular nitrogen roto-vibrational Raman signal. The two signals are characterized by an almost identical overlap function, and therefore the overlap effect cancel out when ratioing the signals. Conversely, particle backscattering coefficient profiles at 1064 nm are obtained through the application of a Klett-modified inversion approach (Di Girolamo *et al.*, 1995, 1999). The specific approach used in the present analysis considers a height-dependent lidar ratio profile and iterative procedure converging to a final particle backscattering profile Di Girolamo et al. (1995, 1995). It is to be specified that potential effects on our measurements associated with the incomplete overlap of the laser beam and field of view are negligible. In fact, the particle backscattering coefficient at both 355 and 532 nm are obtained through the Raman lidar technique, which considers the ratio of the elastic echoes over the corresponding $N_2$ Raman lidar echoes, with the overlap function for the elastic channel and the corresponding $N_2$ Raman channel being substantially identical. Additionally, the elastic backscatter signal at 1064 nm and an additional elastic backscatter signal at 532 nm are collected with two small telescopes, developed around two 50 mm-diameter 200 mm-focal length lenses, with overlap regions not extending above 3-400 m." However, for the purpose of clarity this information is now also partially anticipated in an earlier stage of the paper. In

this regard, the sentence in former line 76, has been modified as follows. "Elastic backscattering echoes from aerosol and molecular species at 355, 532 and 1064 nm, in combination with the Raman scattering echoes from molecular nitrogen, are used to measure the vertical profiles of the aerosol backscattering coefficient at these three wavelengths. More details of the considered approaches are given in section 4."

Line 151: In the equation is missing a bracket (please use square brackets). Moreover, some terms are poorly or not at all described.

We believe the referee is referring here to the equation 5 in line 131 and not in line 151. Here we could not find where a bracket was missing, but we introduced the term $\ln r - \ln r_m$ within square brackets instead of round brackets and we also squared the term within the square brackets ($[\ln r - \ln r_m]^2$). Additionally, we specified that $n(r)$ represents the number of particle within the size interval $dr$, $N_0$ represents the particle integral concentration, $r_m$ represents the median radius of the distribution, $S$ is the standard deviation of the distribution, which is a measure of the particle polydispersity. The portion of sentence following equation 5 has been modified as follows: "where $n(r)=dN/dr$ is the number of particles within the size interval $dr$, with $N(r)$ representing the cumulative particle number distribution for particles larger than $R$, $r_m$ is the median radius of the distribution, $S$ is the standard deviation of the distribution and $N_0$ is the particle integral concentration for the considered mode". The following sentence was also introduced: "$S$ is a measure of the particle polydispersity, with $\ln S$ being equal to 1 for monodisperse particles".

Lines 156-158: the sentence is not clear at all, please rephrase.

There is no sentence starting in line 156 and ending in line 158. If the referee refers to the sentence starting in line 155 and ending in line 157, this sentence has been modified as follows: "Specifically, figure 4 illustrates back-trajectories computed with the NOAA HYSPLIT model of the air masses overpassing the lidar site at 20 UTC on 13 September 2012 at the altitudes of 600 m (red line), 4000 m (blue line) and 6000 m (green line). The trajectories extend back in time for 5 days, thus illustrating the air masses path since 20 UTC on 8 September 2012."

Line 186, again, smaller is not a good choice. Use rather "poorer"

The term "smaller" is not present in line 186. If the referee refers to the term "smaller" present in line 203, this sentence has been modified as follows: "In this case, the agreement between measured and simulated profiles is poorer, but still acceptable up to 3000 m."

Fig. 16 X-axis label is missing

The X-axis label was added, which is "Quadratic sum of deviations at 355 nm and 532 nm".

**References**

Estellés, V., J. A. Martınez-Lozano, and M. P. Utrillas, 2007: Influence of air mass history on the columnar aerosol properties at Valencia, Spain. J. Geophys. Res., 112, D15211, doi:10.1029/2007JD008593.

Man, C.K., and M.Y. Shih, Identification of sources of PM10 aerosols in Hong Kong by wind trajectory analysis, Journal of Aerosol Science, Volume 32, Issue 10, Pages 1213-1223, https://doi.org/10.1016/S0021-8502(01)00052-0, 2001.

Methven, J., M. Evans, P. Simmonds, and G. Spain, 2001: Estimating relationships between air mass origin and chemical composition. J. Geophys. Res., 106, 5005–5019.

Toledano, C., V. E. Cachorro, A. M. De Frutos, B. Torres, And A. Berjon, M. Sorribas, R. S. Stone, Airmass Classification and Analysis of Aerosol Types at El Arenosillo (Spain), Journal Of Applied Meteorology And Climatology, Volume 48, 962-981, doi: 10.1175/2008jamc2006.1, 2009.

**Anonymous Referee #3**

This paper deals with the combined analysis of an optical particle counter deployed on airplane and lidar with the objective of providing aerosol typing at different altitudes.
The topic is sounds and deserve a publication. The paper is also generally well-written and structured, although eventually lack of consistent scientific conclusions about the method proposed.

The paper has been expanded and partially re-written with the aim of providing consistent scientific conclusions concerning the proposed methods and their applicability. The paper has been heavily reshuffled and now both case studies are analyzed with the same approach previously used for the second case study. This approach is not using back-trajectory analysis to make preliminary assumption of the aerosol type, but is using back-trajectory analysis only afterward as a verification tool. In addition to a variety of other integrations throughout the manuscript, the conclusions have been largely expanded proving scientific consistency to the proposed method and the achieved results.

Before recommending the manuscript publication I have several issues that needs to be addressed.

MAJOR CONCERNS

My main concern is about the conclusions obtained in both studies cases when aerosol loads is very low, which is observed in the free troposphere for your study cases. Authors should address this limitation of their methodology by for example doing sensitivity tests using synthetic data. Such analysis will help to better understand the differences you obtain in your analyses.

We agree with the referee that the applicability of the considered methodology in those cases when the aerosol loads is very low, typically in the free troposphere, has to be assessed through a specific sensitivity analysis based on the use of synthetic data. In this regard, synthetic profiles for the particle backscattering coefficient at the three wavelengths have been determined considering a certain range of variability for specific parameters. In what follows, for example, we discuss the sensitivity of the particle backscattering coefficient at 355 nm to variable values of the real part of the refractive index. We considered the second case study, i.e. the time interval 19:43-20:27 UTC on 02 October 2012. Specifically, the left panel of figure 1 below shows the variability of particle backscattering coefficient at 355 nm for a ± 5 % variability of the real part of the refractive index for urban aerosols. These aerosols include a soot and pollution fine mode component, with a the real part of the refractive index of 1.75 (which was varied between 1,66 and 1,84), a water soluble accumulation mode, with a the real part of the refractive index of 1.53 (varied between 1,45 and 1,61), and component and dust-like coarse mode component, also with a the real part of the refractive index of 1.53 (also varied between 1,45 and 1,61). The right panel of figure 1 shows the variability of deviations between measured and simulated particle backscattering profiles at 355 nm. Results in this figure reveal that, as a result of the a ± 5 % variability of the real part of the refractive index for all three modes the deviations between measured and simulated particle backscattering profiles typically increase by approx. 20 % from ~ 10 to ~ 30 %, which is not compromising the correct application of the algorithm. A comparable increase is observed for the quadratic sum of deviations at 355 nm and 532 nm ($\Delta_{tot} = \sqrt{\Delta_{355}^2 + \Delta_{532}^2}$), with values of increase by approx. 20 % from ~ 25 to ~ 45 % (figure 2). In this respect, it is to be pointed out that a ± 5 % variability of the real part of the refractive index, while appearing as a fairly limited variability, may correspond to a substantial change in aerosol composition. In this regard, it is to be underlined that values of real part of the refractive index for most aerosol components are within ± 5 % of the value of Sulphate aerosols (1,45): Sea-salt (1,39) Dust-like (1,53) Mineral (1,53), Water soluble (1.53) and Sulphate (1.45).

[Figure]

Figure 1. left panel: particle backscattering coefficient at 355nm for the time interval 19:43-20:27 UTC on 02 October 2012, variability for a ± 5 % variability of the real part of the refractive index for urban aerosols; right panel: variability of deviations between measured and simulated particle backscattering profiles at 355 nm.

[Figure]

Figure 2: Variability of the quadratic sum of deviations at 355 nm and 532 nm, $\Delta_{tot}$, again for the time interval 19:43-20:27 UTC on 02 October 2012.

The text has been integrated with the introduction of the following text:
"A sensitivity study has been also carried out to assess the sensitivity of the results to changes of specific size and microphysical parameters' values. The sensitivity study reveals that the considered

methodology for aerosol typing is successfully applicable in the altitude region up to 3900 m, as in fact above this altitude the statistical uncertainty affecting the lidar signals is high and this severely reduces the sensitively of the aerosol typing methodology. The sensitivity analysis also reveals that in the lower levels, typically within the boundary layer where aerosol loading is larger, deviations between measured and simulated particle backscattering coefficient at the three wavelengths may vary by up to 20 % as a result of a ± 5 % variability of specific size and microphysical parameters (for example, the real part of the refractive index), which certainly reduces confidence in the aerosol type assessment, but is not compromising its outcome. Based on the results from this study we may conclude that the use of particle backscattering measurements at two wavelengths in combination with OPC measurements allow to get a sufficiently reliable assessment of the aerosol types, which can be verified and refined based on the use of back-trajectory analyses."

It is necessary to improve the methodology section about how you obtain different aerosol types from the fitting procedure you propose.

As also indicated to the other referee, in the revised version of paper we have aligned the analysis of the first and second case studies, using the same approach (the approach formerly applied to the second case study) to both case studies. The methodology section has now been completely rewritten in the direction to include a detail description of how different aerosol types are obtained through the fitting procedure. More specifically, the following new text has been introduced:

[revised manuscript text omitted]

Finally, the possibility of having continental instead of marine aerosol at high levels for the case study on 13th September 2012 has been tested using for the first case study the same methodology considered for the second test case.

It is not clear to me what aerosol parameters you are obtaining.

The aerosol parameters that are obtained through the application of the first approach are the particle concentration $N_{0,i}$ for the three different aerosol modes. Particle concentration $N_0$ is obtained by minimizing differences between the size distribution measured by the OPC and the simulated distribution, while the values of $r_m$ and $S$ are those identified based on literature results. Based on this information, the simulated profiles of the particle backscatter coefficient at 355, 532 and 1064 nm are obtained. This is now better specified in the text (see previous answer).

I would recommend adding references to Table 1.

Former Table 1 has been removed, while former Table 2 (now Table 1) has been modified to include information previously present in former Table 1.Specific references to table 1 have now been introduced in the text.

It is not clear to me the capabilities of BASIL and what you are actually using. You say that they are capable of obtaining independent extinction and backscattering at 355 and 532 nm by Raman technique. But at some points in the manuscript you mention that you use only elastic signals at 532 nm. If this is true, it means that you are using the classical Klett method for obtaining backscattering method that imply constant lidar ratios and thus homogenous aerosol in the entire column. What is then the sense of applying your methodology for aerosol typing? This question is also applicable for backsctttering at 1064 nm.

The system is in principle capable to independently measure extinction and backscattering coefficient profiles at 355 and 532 nm. Extinction coefficient profiles at 355 and 532 nm are obtained from the molecular nitrogen Raman lidar echoes at 386,7 and 607 nm, respectively, through the application of the approach proposed by Ansmann et al. (1990). This approach is based on the application of a derivative operator to the two $N_2$ Raman lidar echoes and provides accurate results only in case the two $N_2$ Raman lidar echoes are characterized by very limited statistical fluctuations. Conversely, backscattering coefficient profiles at 355 and 532 nm are obtained from the elastic echoes at these two wavelengths in combination with the molecular nitrogen Raman lidar echoes at 386,7 and 607 nm, respectively, through the application of the approach by Ansmann et al. (1992). This approach considers the ratio of the elastic echoes over the corresponding $N_2$ Raman lidar echoes. It is to be specified that, for the specific case studies considered in this manuscript, $N_2$ Raman lidar echoes at 386,7 and 607 nm are sufficiently intense, and consequently affected by sufficiently low statistical fluctuations, to allow the determination of particle backscattering coefficient profiles through the Raman technique (Ansmann et al., 1992), but their intensity is not sufficiently high, and consequently statistical fluctuations are too high, to allow the determination of particle extinction coefficient profiles through the application of the above mentioned derivative approach (Ansmann et al., 1990).
The application of the Klett approach is considered only for the determination of the particle backscattering coefficient at 1064 nm. In this regard, however, we need to specify that the approach used in the present analysis refers to a modified version of the traditional Klett approach, proposed by Di Girolamo et al. (1995, 1995), which considers a height-dependent lidar ratio profile and iterative procedure converging to a final profile. Thus, the considered analysis methodology does not imply a homogenous aerosol layer throughout the entire column and its application does not alter its value for aerosol typing. This is now clearly specified in the paper, where the following

sentences have been introduced: "The specific approach used in the present analysis considers a height-dependent lidar ratio profile and iterative procedure converging to a final particle backscattering profile Di Girolamo et al. (1995, 1995)."

The two study cases are not very different. The area is usually affected by Saharan dust transport. The paper will definitely gain more interest if an study case affected by dust transport is included.

While we agree with the referee on the importance and potential impact for this research effort of considering Raman lidar measurements in the presence of Saharan dust particles, we have to specify that simultaneous and co-located measurements of the Raman lidar and the air-craft sensors (the OPC being one of these) are available only during four specific days (i.e. 13 September, 02 and 29 October and 05 November 2012), when flights hours of the French research aircraft ATR42 dedicated to the 7th FP "WALiTemp" project were carried out. Unfortunately, during these days no Saharan dust outbreak event took place and consequently no evidence of Saharan dust layers could be identified in the lidar echoes. Indeed the two considered case studies are not very different, but none of the other potential candidate cases (29 October and 05 November 2012) reveals any more significant peculiarity (on these latter two days, winds are from N-NO and a very limited aerosol loading is observed).

MINOR CONCERNS

Why using HYSPLIT at different altitudes between the two examples? Please unify criterion. There should not be dramatical changes in your interpretation.

We agree with the referee that it was confusing and not logical to consider in the HYSPLIT back-trajectory analysis different altitudes for the two case studies. We modified the paper in the direction to consider the same altitudes for the two case studies. We are presently considering for both case studies the following altitude levels: 600 m, 4000 m and 6000 m.

I agree with the previous referee about the interpretation of marine aerosol at high levels for the case study on 13th September 2012. See my previous comments about the applicability of your methodology for very low aerosol loads.

This point has been already extensively addressed above (see specific replies to referee # 2 in the initial part of the response letter to him/her). Additionally, as we also illustrated in our reply to referee # 2, it is to be specified that during the field campaign a very intense hurricane (hurricane Nadine) took place in the area near Azores, covering a long time period between 10 September and 04 October 2012, with the two selected case studies considered in this paper being on 13 September and 02 October 2012, i.e. within this period. More specifically, the back-trajectory analysis carried out with a Lagrangian model (HYSPLIT) reveals that the air mass overpassing the lidar site at an altitude of 4000 m on 2 October 2012 was overpassing few days earlier (29-30 September 2012) an area approximately 1000 km North of the Nadine hurricane. This information provides some confidence on the possible presence of marine aerosol in the upper troposphere. Nevertheless, we removed from the interpretation of the observations references to the presence of marine aerosols in the upper troposphere.

Lines 30-35: Please provide references.

As requested by the referee, references have been provided in support of the argument that only a limited number of remote sensing techniques can provide vertically resolved measurements of aerosol microphysical properties. Specifically, the following references have been introduced: Bellantone et al., 2008; Granados-Muñoz et al., 2016; Mhawish et al., 2018.

Line 164: 'particles at higher altitudes are internal mixtures of…' It is impossible to know if they are internal or external mixtures with the instrumentation and methodology you used. Please correct.

In the revised version of the paper this sentence has been removed, as well as any reference in the interpretation to the internal mixture of continental polluted aerosols with marine salt. Additionally, we agree with the referee that it is impossible to argue whether observed aerosols are internal or external mixtures based on the instrumentation and methodology considered in the paper. Any reference in the interpretation of the results to internal/external aerosol mixtures has been removed.

Figure 1 needs improvements as in its current form does not provide any information. Why not include flight tracks?

The ideal flight track of the ATR 42 was already present in the previous version of the paper. This was represented by the red line oblong circle, which indicates the footprint of the spirals (hippodromes) up and down around a central location approximately 20 km eastward of the Raman lidar site. In the new revised version of this figure we also included the flight track from the airport to the spiraling position. Additionally, to improve the information content of this figure, we also introduced a latitude-longitude grid. Now the figure caption reads: "ATR42 flight pattern in the frame of the "WaLiTemp" project (red line). The light blue dot represents the position of Montpellier Airport, where the ATR-42 was taking-off and landing, while the red dot represent the position of the Raman lidar BASIL. The red curve represents the footprint of the aircraft pattern, including the positions of the spirals (hippodromes) up and down and the ground track from the airport to the spiraling position. The distance between the lidar site and the flight pattern is approx. 20 km."

What are the effects of incomplete overlap in your methodology?

Effects of incomplete overlap are negligible in our methodology. In fact, the particle backscattering coefficient at both 355 and 532 nm are carried out based on the application of the Raman lidar technique (see more detail above), i.e. considering the ratio of the elastic echoes over the corresponding $N_2$ Raman lidar echoes. The system received was conceived to be compact enough in order to have almost identical overlap functions for the elastic signal at 355 nm and the corresponding $N_2$ Raman lidar signal at 386,7 nm, as well as for the elastic signal at 532 nm and the corresponding $N_2$ Raman lidar signal at 607 nm. For what concerns the particle backscattering coefficient at 1064 nm, it is to be specified (and this is clearly described in the paper in section 2.1, describing the system layout for the Raman lidar) that the corresponding signals are not collected with the large aperture Newtonian telescope, but with a small telescope developed around a 50 mm-diameter 200 mm-focal length lens. This receiver is located within 10 cm from the vertically propagating laser beam and this determines the presence of a very shallow overlap region, not extending above 3-400 m. Same is true for the elastic signals at 532 nm, which are collected with the large aperture Newtonian telescope, but are also collected with a second small telescope (again a 50 mm-diameter 200 mm-focal length lens). This alternative 532 nm signal, used when the $N_2$ Raman lidar signal at 607 nm is not strong enough to allow applying the Raman technique, is characterized by the presence of a very shallow overlap region, as the one of the 1064 nm signal, not extending above 3-400 m. This aspect is now better stressed in the paper, where the following sentences have been introduced in the Methodology section of the paper: "It is to be specified that potential effects on our measurements associated with the incomplete overlap of the laser beam and field of view are negligible. In fact, the particle backscattering coefficient at both 355 and 532 nm are obtained through the Raman lidar technique, which considers the ratio of the elastic echoes over the corresponding $N_2$ Raman lidar echoes, with the overlap function for the elastic channel and the

corresponding N$_2$ Raman channel being substantially identical. Additionally, the elastic backscatter signal at 1064 nm and an additional elastic backscatter signal at 532 nm are collected with two small telescopes, developed around two 50 mm-diameter 200 mm-focal length lenses, with overlap regions not extending above 3-400 m."

Line 325-326. 'decreasing number concentration of the same aerosol type (Continental/Urban) as a result of the progressive attenuation of the underlying convective activity' I do not understand this sentence. Please clarify.

This unclear sentence was removed. The corresponding sentence has been changed as follows: "Above the top of the boundary layer and up to ~2700 m (altitude 2), particle backscatter decreases with altitude".

Conclusions must be improved.

Conclusions have been sensitively improved. The following new text have been introduced:
"The added value of the reported methodology is represented by the possibility to infer the presence of different aerosol types and their microphysical properties based on the use of multi-wavelength Raman lidar measurement from a ground-based system in combination with an independent measurement of the particle concentration profile (in our case we are using the one coming from an optical particle counter mounted onboard an aircraft overpassing the lidar site). This methodology is applicable when sounded particles are spherical or almost spherical, which allows for the Mie scattering theory to be applied for the determination of the particle backscattering coefficient.
The HYSPLIT-NOAA back-trajectory model was used to verify the origin of the sounded aerosol particles. Five different aerosol typologies are considered, i.e. Continental polluted, Clean continental-rural, Urban, Maritime-polluted and Clean-polar, with their size and microphysical properties taken from literature. The approach leads to an assessment of the predominant aerosol component based on the application of a minimization approach applied to the deviations between measured and the simulated particle backscattering profiles at 355 and 532 nm and for first test case study also at 1064 nm, considering all five aerosol typologies.
The application of this approach to the case study on 13 September 2012 suggests the presence of Urban and Maritime aerosols throughout the entire vertical extent of sounded column, except in the altitude region 2600-2900 m and 4300-4500 m ranges, where the presence of a Rural component is likely to be possible. The application of the approach to the case study on 02 October 2012 reveals that Continental/Urban aerosols are likely to be the predominant components up to 1600 m, while the two distinct aerosol layers located in the altitude regions 2700-3600 m (with max. at 3000 m) and 3600-4600 m (with max. at 4000 m) are identified to likely consists of Continental/Urban and/or Marine polluted aerosols, respectively. The correctness of the results has been verified based on the application of the HYSPLIT-NOAA back-trajectory model, with the analysis extend backing in time for 5 days and allowing to assess the origin of the sounded aerosol particles.
Finally, a sensitivity study has been carried out to assess the sensitivity of the aerosol typing approach to varying size and microphysical parameters. The study reveals that the reported approach is successfully applicable in the altitude region up to
~ 4 km, while above this altitude the sensitively of the approach is sensitively reduced by the high statistical uncertainty affecting lidar signals. The sensitivity study also reveals that the within the boundary layer deviations between measured and simulated particle backscattering coefficient at 355, 532 and 1064 nm may vary up to 20 % as a result of a ± 5 % variability of specific size and microphysical parameters. Such results reveal that application of the reported approach, based on the use of particle backscattering measurements at two wavelengths in combination with OPC measurements, allow to get a sufficiently reliable assessment of aerosol typing.".

What is the added value of your methodology?

The added value of our methodology is represented by the possibility to infer the presence of different aerosol types using two or three wavelength Raman lidar measurement from a ground-based system in combination with an independent profile of particle concentration (in our case we are using the one coming from an optical particle counter mounted onboard an aircraft overpassing the lidar site). This aspect is now clearly specified in the Conclusions, where the following sentence has been introduced: "The added value of the reported methodology is represented by the possibility to infer the presence of different aerosol types and their microphysical properties based on the use of multi-wavelength Raman lidar measurement from a ground-based system in combination with an independent measurement of the particle concentration profile (in our case we are using the one coming from an optical particle counter mounted onboard an aircraft overpassing the lidar  site)."

Under which circunstances is applicable?

The circumstances under which the methodology is applicable are those when sounded particles are spherical or almost spherical, which allows for the Mie scattering theory to be applied for the determination of the particle backscattering coefficient. This aspect is now clearly specified in the Conclusions, where the following sentence has been introduced: "This methodology is applicable when sounded particles are spherical or almost spherical, which allows for the Mie scattering theory to be applied for the determination of the particle backscattering coefficient".

Which aerosol properties are you providing?

The proposed approach is not aimed at determining specific aerosol properties, but instead it is aimed at inferring the different aerosol types present in the vertical column above the lidar site based on the use of particle backscattering measurements at two wavelengths from the Raman lidar in combination with OPC measurements. This aspect is now clearly specified in different parts of the paper. In the Abstract, where the following sentence is present: "The reported good agreement between measured and simulated multi-wavelength particle backscatter profiles testifies the ability of multi-wavelength Raman lidar systems to infer aerosol types at different altitudes." In the Introduction, where the following sentence has been introduced: "Specifically, lidar systems with aerosol measurement capability are characterized by high accuracies and temporal/vertical resolutions, which makes them particularly suited for aerosol typing applications." Further down in the Introduction, where the following sentence has been introduced: "In the present manuscript, measurements carried out by BASIL are illustrated with the purpose to characterize atmospheric aerosol optical properties. These measurements, in combination with in-situ measurements from an airborne optical particle counter and the application of a Mie scattering code, are used to infer aerosol types." In the Methodology section, where the following sentence has been introduced: "A modified version of the approach defined by Di Iorio et al. (2003) was applied in order to determine the sounded aerosol typology. This approach is based on the minimization of the relative deviation between the measured and the simulated particle backscattering coefficient …". In section 5.2, where the following sentence has been introduced: "Based on the results from this study we may conclude that the use of particle backscattering measurements at two wavelengths in combination with OPC measurements allow to get a sufficiently reliable assessment of the aerosol types, which can be verified and refined based on the use of back-trajectory analyses." Finally, in the Summary, where the following sentence has been introduced: "Such results reveal that application of the reported approach, based on the use of particle backscattering measurements at two wavelengths in combination with OPC measurements, allow to get a sufficiently reliable assessment of aerosol typing."